# Drivers of drought-induced shifts in the water balance through a Budyko approach

Tessa Maurer[1,2], Francesco Avanzi[3], Steven D. Glaser[2], and Roger C. Bales[2,4]

[1]Blue Forest Conservation, Sacramento, CA, USA
[2]Department of Civil and Environmental Engineering, University of California, Berkeley, CA 94720, USA
[3]CIMA Research Foundation, via Armando Magliotto 2, 17100, Savona, Italy
[4]Sierra Nevada Research Institute, University of California, Merced, CA, USA

**Correspondence:** Tessa Maurer (tmaurer@berkeley.edu)

**Abstract.** An inconsistent relationship between precipitation and runoff has been observed between drought and non-drought periods, with less runoff usually observed during droughts than would be expected based solely on precipitation deficit. Predictability of these shifts in the precipitation-runoff relationship is still challenging, largely because the underlying hydrologic mechanisms are poorly constrained. Using 30 years of data for 14 basins in California, we show how the Budyko framework can be leveraged to decompose shifts in precipitation versus runoff during droughts into "regime" shifts, which result from changes in the aridity index along the same Budyko curve, and "partitioning shifts", which imply a change in the Budyko parameter $\omega$ and thus to the relationship among water balance components that governs partitioning of available water. Regime shifts are primarily due to measurable interannual changes in precipitation or temperature, making them predictable based on drought conditions. Partitioning shifts involve further nonlinear and indirect catchment feedbacks to drought conditions and are thus harder to predict a priori. We show that regime shifts dominate changes in absolute runoff during droughts, but that gains or losses due to partitioning shifts are still significant. Low aridity, high baseflow, a shift from snow to rain, and resilience of high-elevation runoff correlate to higher annual runoff during droughts than would be predicted by the precipitation-runoff ratio during non-drought years. Differentiating between these shifts in the precipitation-runoff relationship using a Budyko approach will help water resources managers, particularly in arid, drought-prone regions, to better project runoff magnitudes during droughts based on available climate data and, furthermore, understand under what circumstances and to what extent their forecasts may be less reliable due to nonlinear basin-climate feedbacks.

## 1 Introduction

Droughts can threaten human and natural systems worldwide, accounting for more than 50% of all natural hazard deaths over the course of the 20th and early 21st centuries (Van Loon, 2015; Maskey and Trambauer, 2015). As baseline water stress intensifies globally due to growing populations and land-use changes (Hofste et al., 2019), the impact of meteorological and hydrologic droughts may become more severe (Masih et al., 2014). In Mediterranean climates where the bulk of precipitation falls during winter, while summers are dry, droughts exacerbate already significant water-management challenges, as these basins typically rely on intricate systems of natural and built water storage to maintain water supply across regularly occurring

seasonal and multi-year dry periods (He et al., 2017; Woodhouse et al., 2010). The need to adequately understand and predict the water balance implications of droughts is becoming more acute as climate change makes basins susceptible to more severe and prolonged droughts (Dai, 2013; Trenberth et al., 2014; Woodhouse et al., 2010).

Chief among these water balance implications is the relationship between precipitation and runoff, which has been shown to be impacted inconsistently by drought depending on the basin. In some areas, a meteorological drought causing reduced precipitation results in a predictable and commensurate decrease in runoff (Tian et al., 2020; Avanzi et al., 2020; Saft et al., 2016; Coron et al., 2012; Vaze et al., 2010). In other words, a consistent precipitation-runoff relationship applies across across drought and non-drought periods. However, this pattern does not hold universally; in some areas where the precipitation-runoff relationship changes during droughts, observed runoff is less than would be predicted using non-drought relationships. These drought-induced shifts have been observed in basins around the world (e.g., Saft et al., 2016; Potter et al., 2011; Tian et al., 2020; Avanzi et al., 2020; Alvarez-Garreton et al., 2021), compounding water shortages for municipal, industrial, and agricultural systems.

Despite documentation of these shifts and their implications for human water supply, it is not fully understood why only some basins show a change in hydrologic functioning during droughts and, furthermore, what causes those shifts in places they are observed (Bales et al., 2018; Peterson et al., 2021). Prior studies, including Saft et al. (2016), Potter et al. (2011), and Avanzi et al. (2020), used linear-regression-based approaches to identify factors associated with drought-induced changes to the precipitation-runoff relationship in Australia and California. Tian et al. (2020) used a multivariate generalized additive model to address a similar question in a slightly wetter, monsoon-dominated region in China. These studies were able to identify factors associated with changes to the water balance during drought, including aridity, rainfall seasonality, vegetation feedbacks, and catchment elevation, but it is still not clear whether changes to the precipitation-runoff relationship can be expected for a given basin and drought and, if so, to what extent.

Furthermore, linearized models may not fully capture the well-documented nonlinearities in the relationships between water balance components, such as that between runoff and storage (Kirchner, 2009) or ET and storage (Avanzi et al., 2020). While these nonlinearities between water balance components almost certainly contribute to changes in water partitioning during droughts, it is not always clear which nonlinear relationship(s) are the dominant driver of observed changes for a given basin and drought period, nor what physical mechanisms are behind them. For example, several catchment feedbacks may create nonlinearities and/or hysteresis in the ET-storage relationship during drought relative to wet periods (Avanzi et al., 2020), but these feedbacks are dependent on drought duration and severity. Soil water storage can decouple ET from precipitation by allowing vegetation to withstand periods of mild to moderate drought (Bales et al., 2018; Oroza et al., 2018; Hahm et al., 2019b; Tague and Grant, 2009), while vegetation stress responses to depleted subsurface storage, including stomatal closure (Avanzi et al., 2020; Goulden and Bales, 2019) and tree die-offs (Bales et al., 2018), may occur during more severe, pro-longed droughts. Other mechanisms that can drive nonlinearities between ET and storage include climate-induced changes like increasing temperatures, which can increase ET in areas with sufficient water (Teuling et al., 2013; Mastrotheodoros et al., 2020) and influence precipitation phase and the elevation of the snow line in basins with significant snowfall (Zhang et al.,

2017). These changes, in turn, influence the timing of available water (Rungee et al., 2019; Avanzi et al., 2020) and the spatial distribution of runoff production in the basin (Avanzi et al., 2020; Bales et al., 2018).

Thus, there is an opportunity to revisit the question of drought-based shifts in the water balance through an explicitly non-linear approach. The Budyko hypothesis (Budyko, 1974) is a conceptual water balance model that has been used in numerous catchments around the world to characterize the long-term water balance as a trade-off between supply (precipitation) and demand (PET; e.g., Li et al., 2013; Zhang et al., 2008, 2001; Greve et al., 2016; Moussa and Lhomme, 2016; Shen et al., 2017; O'Grady et al., 2011; Gnann et al., 2019). As a conceptual model, the Budyko framework can provide a macroscale

understanding of the relationship between water balance components across a catchment, while minimizing the need for high-resolution data or large parameter sets (Hrachowitz and Clark, 2017). Since it accounts for ET, it allows for consideration of the nonlinearity in the P-Q relationship across a variety of climatic conditions. The Budyko approach has been leveraged to examine the water balance impacts of general climatic changes (Li et al., 2019; Wang and Alimohammadi, 2012), vegetation, (Zhang et al., 2016; Ning et al., 2019; Oudin et al., 2008), and land-use changes or other human activity (Liu et al., 2017; Shen

et al., 2017), but its application to drought impacts specifically has been limited (see, e.g., Huang et al., 2017; Graf et al., 2020).

    A number of studies have employed some variation of decomposing movement in the Budyko space to analyze contributing factors to a particular phenomenon, such as controls on the water balance beyond radiation and water availability (Williams et al., 2012), drivers of change in forest evapotranspiration (Jaramillo et al., 2018), and changes to runoff (Zhang et al., 2019; Wang and Hejazi, 2011). The direction of movement indicates the factors driving change in the basin. For example, Jaramillo

et al. (2018) analyze movement along each axis in the Budyko space to differentiate between climatic effects (movement induced by changes to the aridity index) versus other "residual" effects.

    In this paper, we hypothesize that the Budyko framework, as an explicitly nonlinear approach that includes ET, may provide additional insights in water balance shifts during droughts compared to linearized methods. Following a decomposition approach, we apply the Budyko framework to analyze drought-induced shifts in the precipitation-runoff relationship, charac-

terizing the water balance across three droughts in 14 basins in the Sierra Nevada. We distinguish "regime" shifts, which result from changes in the aridity index along the same Budyko curve, from "partitioning" shifts, which imply a change in the Budyko calibration parameter and thus to the relationships between evaporative demand, precipitation, and ET that govern partitioning of available water. We aim to address the following questions: 1) How can a nonlinear framework be used to identify changes in the precipitation-runoff relationship during droughts?; 2) Within the Budyko framework, how are changes in ET and runoff

due to drought-induced climate variation distinguished from changes due to a shift in water balance relationships? What is the quantified impact of these changes?; and 3) How do quantified changes due to shifts in water balance relationships correlate to known basin drought response mechanisms?

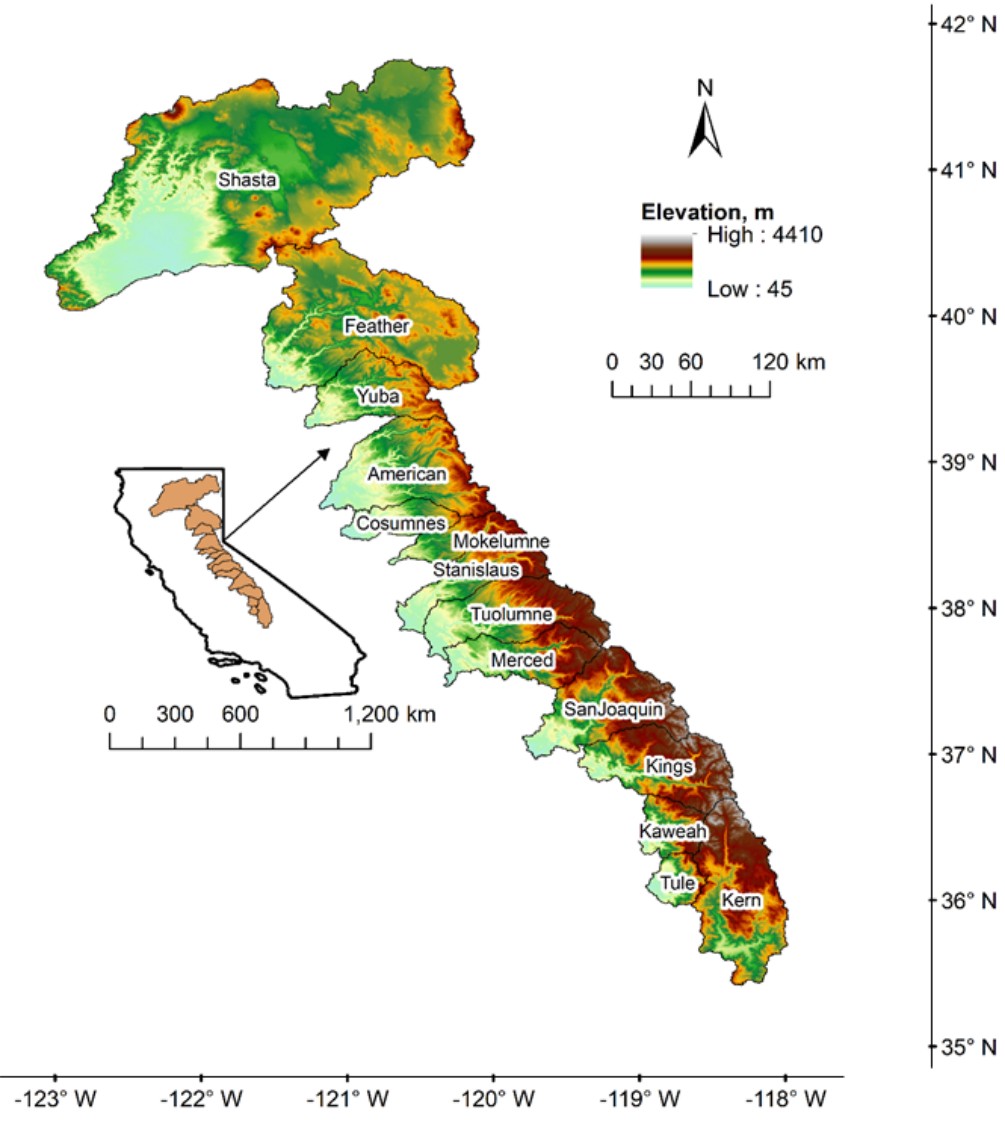

**Figure 1.** Map of California, indicating extent of the river basins used in this study. The northern Sierra extends from the Shasta to Cosumnes, the central Sierra from the Mokelumne to the Merced, and the southern Sierra from the San Joaquin to the Kern. Elevation was derived from U.S. National Elevation Database Digital Elevation Models (EROS Data Center, 1999).

## 2 Methods

### 2.1 Study area

Our study area comprises the 14 major river basins draining into the Sacramento-San Joaquin Valley of California (Fig. 1). All basins in the study area have a Mediterranean climate, with seasonal precipitation that falls largely between October and May. The wet season is offset from the peak growing period, which occurs in the warmer summer months. Most basins have headwaters on the eastern edge, with elevations decreasing smoothly to the west. The exceptions are the Shasta, which has headwaters to the east, north, and far western edges and drains to the south; the Feather, the eastern two-thirds of which are lower and rain-shadowed; and the Kern, which has headwaters in the northern portion of the basin and drains to the south. Elevations generally increase from north to south in the Sierra Nevada, from an average elevation of 1530 m in the Feather to 2200 m in the Kern. Shasta has a high peak elevation (4300 m), but little surface area above 2400 m. For ease of reference, we refer to all study basins collectively as the Sierra Nevada. The northern basins or northern Sierra Nevada includes the Shasta, Feather, Yuba, American, and Cosumnes basins; the central Sierra Nevada includes the Mokelumne, Stanislaus, Tuolumne, and Merced; and the southern Sierra Nevada includes the San Joaquin, Kings, Kaweah, Tule, and Kern.

### 2.2 Data

We used gridded data products of precipitation, temperature, and evapotranspiration to estimate water balance components for this study. By using these products, we aimed to assess water balance changes without determining any inputs residually, since doing so would relegate all uncertainties in the data to a single water balance component. Here, we briefly discuss the methods used to create these products and the margin of uncertainty in each.

Precipitation (P) and minimum and maximum temperature on the daily timestep were obtained from the Parameter-elevation Regressions on Independent Slopes Model (PRISM; Daly et al., 2008), a widely used precipitation dataset for montane regions of the Western U.S. (see, e.g., Bolger et al., 2011; Abatzoglou et al., 2009; Ackerly et al., 2010; Raleigh and Lundquist, 2012; Ishida et al., 2017). PRISM spatial maps are created based on a regression between digital elevation models (DEM) and a large collection of ground-based precipitation and temperature data, including from the National Weather Service Cooperative Observer Program and Weather Bureau Army Navy stations; U.S. Department of Agriculture National Resource Conservation Service Snow Telemetry (SNOTEL) and snow courses; U.S. Department of Agriculture Forest Service and Bureau of Land Management Remote Automatic Weather Stations; and California Data Exchange Center (CDEC) stations. Stations are weighted by a variety of factors, including clustering with other stations, distance to pixel, elevation, coastal proximity, and topographic facet. After initial values have been calculated for each pixel, maps are subject to final steps to ensure spatial consistency, such as bound checks on vertical gradients between neighboring cells. PRISM maps are generally regarded as the highest-quality gridded precipitation dataset for the Western U.S., with a monthly mean absolute error of 4.7 to 12.6 mm and a potential annual error of $\pm 98.2$ mm (Daly et al., 2008). However, it is well-established that precipitation uncertainty is high in steep, variable terrain with few ground-based measurements, which includes the montane regions of California (Lundquist et al., 2015; Yang et al., 2009; Rasmussen et al., 2012; Avanzi et al., 2021). This is due to a variety of factors, including under-

catch from wind, wetting loss, evaporation, and trace precipitation at precipitation gauges (Yang et al., 1999) and uncertainty in measuring solid precipitation in areas that receive snow (Rasmussen et al., 2012). As a result, it is common practice to adjust for errors in gauge measurements, including those on which PRISM is based (see, e.g., Allerup et al., 2000; Bales et al., 2009; Ma et al., 2015; Mernild et al., 2015). To combat these data quality challenges on the basin scale, annual precipitation data

from PRISM were adjusted by the long-term average residual of $P - ET - Q$ so total basin storage over the period of record was zero. The adjustment procedure allows for reduction of systemic bias in the precipitation data without assuming that all data uncertainty rests in a specific water balance component and is predicated on the assumption that long-term storage in the basin is stable. The procedure was as follows: using the annual, basin-wide values for precipitation, evapotranspiration, and full natural flow, we calculated the residual of $P - ET - Q$. This value represents the annual change in subsurface and deep ground-

water storage in the basin (note that this value is not the same as $\Delta S$ described in Sect. 2.4 and used as part of the extended Budyko framework, since $\Delta S$ represents only plant-accessible subsurface water). Next, we calculated the average of these annual residuals, which represents the adjustment factor. This value was subtracted from the annual precipitation, yielding the precipitation values used in this study. Adjustment factors ranged between 2.35 mm (0.2% of long-term average precipitation) in the Stanislaus to 85.7 mm (8.6%) in the Shasta basin (full list of adjustment factors is reported in the Supplement).

Potential evapotranspiration (PET) was calculated with the Hamon method (Hamon, 1963) on a daily, pixel-by-pixel basis using gridded mean daily PRISM temperatures. The Hamon method has been previously applied in the Sierra Nevada (e.g., Rungee et al., 2019) and was selected because it depends only on spatial temperature data; radiation and energy-balance measurements that would allow for use of other PET calculation methods are limited in the study area and could thus introduce greater uncertainty if interpolated. Prior work in similar regions suggests the Hamon method may overestimate change in PET

during droughts relative to energy-based methods like Penman-Monteith (Zhou et al., 2020), but we expect this effect to fall within the range of uncertainty in the underlying data. PRISM data (both precipitation and temperature) have a pixel size of 800 m and were downscaled using a nearest-neighbor algorithm to match the 30-m pixel size of the ET data.

        ET datasets were available on an annual (water year) basis with a pixel size of 30 m, calculated for the Sierra Nevada following the method presented and validated in Roche et al. (2020). This dataset was based on an empirical relationship

between ET derived from eddy covariance flux towers in California and local normalized difference vegetation index (NDVI) and precipitation values. Gridded ET is then calculated based on LandSat-derived NDVI and PRISM precipitation grids (Roche et al., 2020). This existing and well-established method follows extensive prior work in regions with similar climates and topography that found eddy covariance to be an accurate method for measuring evapotranspiration (Rana and Katerji, 2000; Wilson and Baldocchi, 2000; Wang et al., 2015). Additionally, statistical approaches have been identified as the most suitable

option for modeling ET in the Sierra Nevada region (Goulden et al., 2012). In the regression, NDVI accounts for variations in land cover and vegetation type, which has been shown to have a strong relationship with ET in semi-arid landscapes (Roche et al., 2020; Groeneveld et al., 2007). Including precipitation as a predictor improves ET estimates, especially in the large portions of the northern Sierra Nevada are significantly wet in winter (see Roche et al. (2020) for a full discussion). Since the ET maps were developed on an annual basis and there is no permanent snow cover in these regions, precipitation phase (rain versus

snow) was not considered in the regression. However, it is important to note that by including precipitation, the ET dataset is

not perfectly independent from the precipitation values used in this study. Previous work has addressed this uncertainty and found that estimates of the four water balance components tally with expectations and existing literature (Avanzi et al., 2020; Rungee et al., 2019; Goulden and Bales, 2014; Goulden et al., 2012; Roche et al., 2018). As with precipitation, uncertainties in the ET dataset are related to both the underlying ground-based data as well as the interpolation method. Roche et al. (2020) estimate modeling uncertainty to be between 10-20% for a given pixel; absent a systematic bias in the data, the aggregate basin-scale ET estimate should be lower. The dataset has also shown good performance in closing the water balance (again, see Roche et al., 2020).

Finally, runoff (Q) was obtained in the form of monthly reconstructed unimpaired flow values at the outlet of each river basin from the California Data Exchange Center (http://cdec.water.ca.gov/index.html); see Supplement for the gauges used. Because almost all major rivers in California are dammed at the basin outlet, reconstructed flows are the only viable estimate of Q available. These full-natural flow (FNF) values were calculated by the California Department of Water Resources, starting with measured impaired streamflow or estimated change in reservoir storage. Reservoir evaporation, basin water exports, and irrigation diversions are added, while basin imports and irrigation return flows are subtracted. Specific adjustments may differ in each basin based on the type of human intervention, quality measured data on the impact of interventions, and information on historical flow regimes vary across basins (Ejeta et al., 2007). While these values are an imperfect substitute for true runoff values, most of the uncertainty in FNF values is related to evapotranspiration from overfull banks and natural wetlands (Huang and Kadir, 2016). This is expected to more heavily impact flows through the Central Valley floor and outflows through the Sacramento-San Joaquin Delta, downstream of outlets of the headwater basins used in this study. Since headwater catchments are relatively undeveloped upstream of the reservoirs at the basin outlets, the full-natural flows are expected to be similar to natural conditions (Huang and Kadir, 2016). This assumption has been validated in prior studies for certain headwater basins in California comparing FNF to P-ET; see, for example, Bales et al. (2018); Roche et al. (2020). FNF in place of runoff has been leveraged extensively in the literature on California (e.g. Guan et al., 2016; Ejeta, 2013; Brown and Bauer, 2010; He et al., 2017; Dettinger and Cayan, 2003; Zeff et al., 2021), and similar substitutions have also been used in other regions (e.g. Avanzi et al., 2021). A comprehensive assessment of the uncertainty in FNF values and the implications of its substitution for runoff is outside the scope of the present study, but we note that these uncertainties may impact the calculation of plant-available water storage (see Sect. 2.3) and thus the estimation of overall available water in the basins. In addition, uncertainties may impact estimations of baseflow as presented in Sect. 3.3.2, as these values were also derived from FNF.

Raster data were binned to two spatial scales we considered in this study: basin-wide and by 100 m elevation bands. All data were obtained for water years 1985–2018 and aggregated from their original timesteps to the annual (water year) timescale. (The water year in California runs from October first through September 30$^{th}$ and is referred to by the latter of the two calendar years that it spans.)

## 2.3 Extended Budyko framework

The original Budyko formulation conceived of the water balance as a trade-off between supply, in the form of water from precipitation, and demand, in other words, potential evapotranspiration. Their mutual availability determines the partition of

water between evapotranspiration and runoff. The aridity index, $PET/P$, is plotted against the fraction of precipitation that goes to ET (evaporative index, $ET/P$). An aridity index less than one indicates an energy-limited area, where vegetation productivity is limited by potential evapotranspiration, while an aridity index greater than one indicates a water-limited area, where water availability is the limiting factor.

The original formulation assumes precipitation as the only form of water supply, meaning it was applicable strictly to natural basins (e.g. without water transfers from built infrastructure) and conditions where change in basin water storage was assumed to negligible (Du et al., 2016; Pike, 1964). Usually, this meant that the framework was applied to the long-term (i.e., 10+ years) water balance, a timescale over which change in storage could be assumed to average out to zero, but some early formulations also looked at the annual timescale in areas where catchment function did not depend heavily on storage (e.g. Pike, 1964). If a basin or application does not meet these original criteria, the framework must be modified to remain consistent with the original assumption of supply and demand. In the headwater regions considered in this study, water transfers into the basin are not significant (Ejeta et al., 2007), but change in soil storage on an annual basis may be, particularly during drought years (Bales et al., 2018). Several existing studies have proposed ways of modifying the original Budyko framework to account for soil storage change (e.g. Zhang et al., 2008; Wang, 2012; Chen et al., 2013; Du et al., 2016). Based on available data for our study area, we adopted the approach of Du et al. (2016), who introduced an "extended" Budyko framework in which precipitation values are adjusted to include plant-accessible soil storage change, essentially expanding the available water supply ($P - \Delta S$; Fig. 2a). This method was validated in arid, headwater montane regions similar to those examined in this study (Du et al., 2016).

In Du et al. (2016)'s method, annual soil storage is estimated using a conceptual mass-conservation approach, the *abcd* model. We chose an independent model for determining $\Delta S$ over taking the residual of $P - ET - Q$ for a couple of reasons. First, this approach allowed $\Delta S$ to be calculated independently from the other water balance components, whereas in taking the residual, all uncertainty from the spatial data sets would be arbitrarily concentrated in one water balance component. Furthermore, it allowed soil storage (assumed to be accessible by plants and thus a potential supplement to precipitation for evapotranspiration demand) to be separated from deep groundwater that would be included in the residual of $P - ET - Q$ but does not actually augment the available water supply for evaporative demand.

The *abcd* model is an explicit water balance model developed by Thomas (1981) that provides estimates of direct and indirect runoff, soil water and groundwater storage, and actual ET, calibrated to streamflow at the basin outlet. This allowed for isolation of the change in plant-accessible soil water storage from deep subsurface storage changes. The *abcd* model assumes that "ET opportunity" ($Y_t$) is a function of available water ($W_t$). $Y_t$ is the sum of actual ET over the timestep ($t$) and soil water storage at the end of the timestep, and $W_t$ is the sum of precipitation over the timestep and soil moisture at the beginning of the time step (Eq. (1)).

$$Y_t(W_t) = \frac{W_t + b}{2a} - \sqrt{(\frac{W_t + b}{2a})^2 - \frac{W_t b}{a}}$$

(1)

This relationship between the ET opportunity and available water is parameterized by $a$ (0~1), representing the tendency for runoff to occur before soil is saturated, and $b$, the maximum ET opportunity. The rate that ET occurs from soil storage is assumed to be proportional to the ET opportunity, and thus soil storage is also a function of $b$ (Eq. (2)).

$$S_t = Y_t \exp\left(-ET_{0t}/b\right) \qquad (2)$$

where $ET_{0t}$ is the initial ET at time $t$ and $Y_t$ is calculated in Eq. (1). Streamflow is the difference between $W_t$ and $Y_t$. The remaining two parameters in the model, $c$ and $d$, control the partitioning of direct runoff from groundwater recharge and discharge. However, since we are interested only in change in soil storage, the last two parameters and related calculations were not used in this study. The *abcd* model was calibrated to runoff (FNF) at the basin outlet, as suggested by Thomas (1981), and was performed across all water years (drought and non-drought) in the record simultaneously. (Though calibrating to ET was also an option, this would have undermined our aim to estimate $\Delta S$ in a way that was decoupled from the other water balance components.) The Kling-Gupta Efficiency (Kling et al., 2012; Gupta et al., 2009) was used as the objective function. Note that using one year of model spin-up for the *abcd* model and calculating change in storage eliminates 2 years from the period of record. For full details of the extended Budyko model, see Du et al. (2016); more information on the *abcd* model, see Wang and Tang (2014). Results of the *abcd* calibration are presented in the Supplement.

Various mathematical models exist to represent data plotted in a Budyko framework; one of the most versatile is the Fu equation, in which the ET fraction of available water (evaporative index) is a function of the aridity index ($PET/P$) and the parameter $\omega$, a constant of integration (Fu, 1981; Zhang et al., 2004). The Fu model, modified for the extended Budyko framework following Du et al. (2016), is given in Eq. (3). We note that the symbol $\omega$ is used here for the constant of integration to be consistent with Du et al. (2016), but, strictly speaking, represents different parameter in the extended form than in the original Fu equation.

$$\frac{ET}{P - \Delta S} = 1 + \frac{PET}{P - \Delta S} - [1 + (\frac{PET}{P - \Delta S})^\omega]^{1/\omega} \qquad (3)$$

The value of $\omega$ will determine how close or far from the theoretical limit lines the data fall; the higher $\omega$, the closer the curve comes to the energy and water limit lines. Thus, for a given $\frac{PET}{P - \Delta S}$ value, $\omega$ reflects the partitioning of available water between ET and runoff (Fig. 2a). The physical meaning of $\omega$ has been connected to various basin characteristics, including vegetation coverage type and density, average slope, and relative soil infiltration capacity (Zhang et al., 2001, 2016; Yang et al., 2007; Jaramillo et al., 2018) as well as climate characteristics such as the seasonal offset between peak precipitation and potential evapotranspiration (Ning et al., 2019). In the context of droughts, changes to the water balance can occur in one of two ways: 1) data can shift along the same curve, changing water balance components due to changes in the water or energy limitations

and 2) data can shift to a new curve with a different $\omega$ value (Fig. 2b). We refer to the former as a regime shift, since the basin becomes more or less energy or water limited, and to the latter as a partitioning shift. Note that the terms "regime" and "partitioning" reflect the current application to droughts, but mirror existing vocabulary from other Budyko decompositions, such as "climatic" and "residual" changes, respectively, in Jaramillo et al. (2018).

For each basin in our study area, we calibrated the Fu equation twice, once for drought years and another for non-drought years, allowing us to assess the changes due to one factor or the other and the implications for ET and runoff. The difference between the two $\omega$ values indicates the direction and intensity of the partitioning shift. In order to understand the effect of the two shift types on ET and runoff, we first calculated the hypothetical drought evaporative indices that would have been seen if only a regime shift had occurred (no change in $\omega$; see "+" data points in Fig. 2b). This was by applying Eq. (3) to the annual observed drought values of $\frac{PET}{P-\Delta S}$ and the non-drought $\omega$. We were then able to compare the hypothetical values to the non-drought values (black circles in Fig. 2b). These two sets of data points were converted to absolute values of ET and runoff based on annual precipitation and change in storage values; the difference between their averages was the impact due to a regime shift. To calculate the impact due to partitioning shifts ("×" data points in Fig. 2b), we subtracted the regime shift impacts from the total observed impacts (inverted triangle data points).

## 2.4 Identifying mechanisms of water balance shifts

The shifts in the partitioning of available water can be related to feedback mechanisms between climatic conditions and catchment characteristics that either exacerbate or mitigate drought (Bales et al., 2018; Teuling et al., 2013; Avanzi et al., 2020). (In this study, "exacerbation" and "mitigation" are used with respect to runoff). We examined four basin characteristics and responses to drought that may relate to observed shifts in the precipitation-runoff relationship. These are mechanisms that have previously been associated with drought-induced shifts in the water balance: 1) amount of plant-accessible storage (here, the value estimated as $\Delta S$; Avanzi et al., 2020; Rungee et al., 2019; Oroza et al., 2018; Hahm et al., 2019a); 2) timing of water availability, which is related to precipitation phase (Avanzi et al., 2020; Rungee et al., 2019; Berghuijs et al., 2014); 3) catchment aridity, which has been correlated with sensitivity to interannual changes in precipitation and departures from the historic mean precipitation (Berghuijs et al., 2014; Saft et al., 2016; Tian et al., 2020); and 4) high-elevation runoff, related to basin spatial heterogeneity that can serve to mitigate drought (Bales et al., 2018). Since not all of these mechanisms are directly measured across the Sierra, we use proxies to estimate their effects. Available soil water storage is estimated from average dry-season flow (July–September) as a proxy. Due to the highly seasonal precipitation in the Sierra Nevada, flow during this period almost exclusively reflects outflow from storage rather than surface runoff. Changes to timing of water availability during drought was estimated by looking at changes to precipitation phase (rain versus snow; Avanzi et al., 2020; Rungee et al., 2019). Phase was estimated using a single-threshold temperature index method on a per-pixel basis. For each day, precipitation in pixels with an average temperature of 1 °C or above was assumed to be rain; otherwise it was assumed to be snow (Berghuijs et al., 2014). Catchment aridity ($PET/P$) was calculated directly, not including soil storage in order to isolate the effects of climate, and averaged over the study period. Finally, high-elevation runoff was estimated as the average annual precipitation

minus ET for elevations above 2000 m. This was compared to the area-normalized annual flow at the basin outlet to estimate the proportion of annual runoff from high elevations.

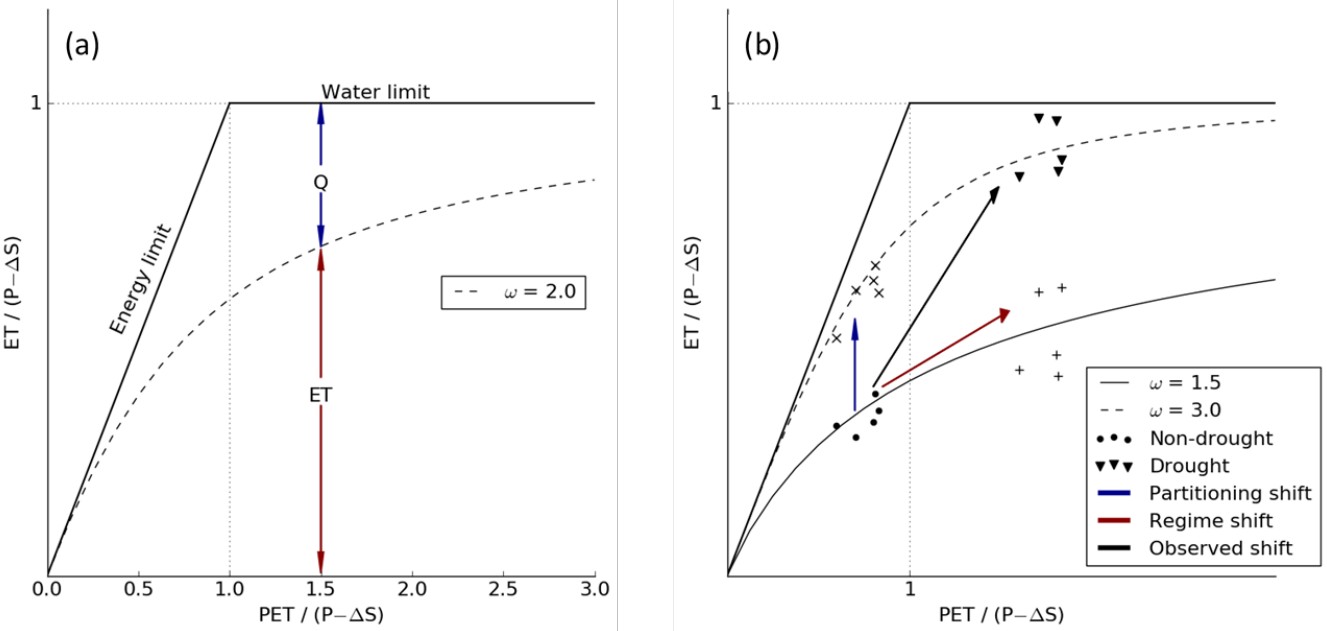

**Figure 2.** (a) Conceptual plot of extended Budyko framework, illustrating how the calibrated Fu equation dictates partitioning of available water and (b) conceptual illustration of drought-induced water balance shifts. ET is evapotranspiration, P is precipitation, $\Delta S$ is change in plant-accessible soil storage, and Q is runoff.

## 3  Results

### 3.1  Drought characterization

The period of record of the available data covers three drought periods, as defined by the State of California (see https://water.ca.gov/Water-Basics/Drought; accessed 29 July 2020): 1987–1992, 2007–2009, and 2012–2016. These droughts are referred to hereafter by the decade in which they ended (1990s, 2000s, and 2010s drought respectively). Average conditions varied across basins and droughts (Fig. 3). Average maximum daily temperature shows no significant change between droughts in the northern basins (average increase of 0.21 °C when all drought years are compared to all non-drought years), but droughts in the central and southern Sierra basins are progressively warmer (average increases of 0.94 and 1.46 °C, respectively). In contrast, average minimum daily temperature shows increases across all droughts and basins (increases of 1.62, 1.88, and 2.11 °C for the northern, central, and southern basins respectively). Average precipitation during the droughts decreases from north to south across the Sierra Nevada, reflecting similar variability in long-term average conditions (average annual precipitation across the period of record was 1245, 1122, and 799 mm in the northern, central, and southern Sierra, respectively). In the

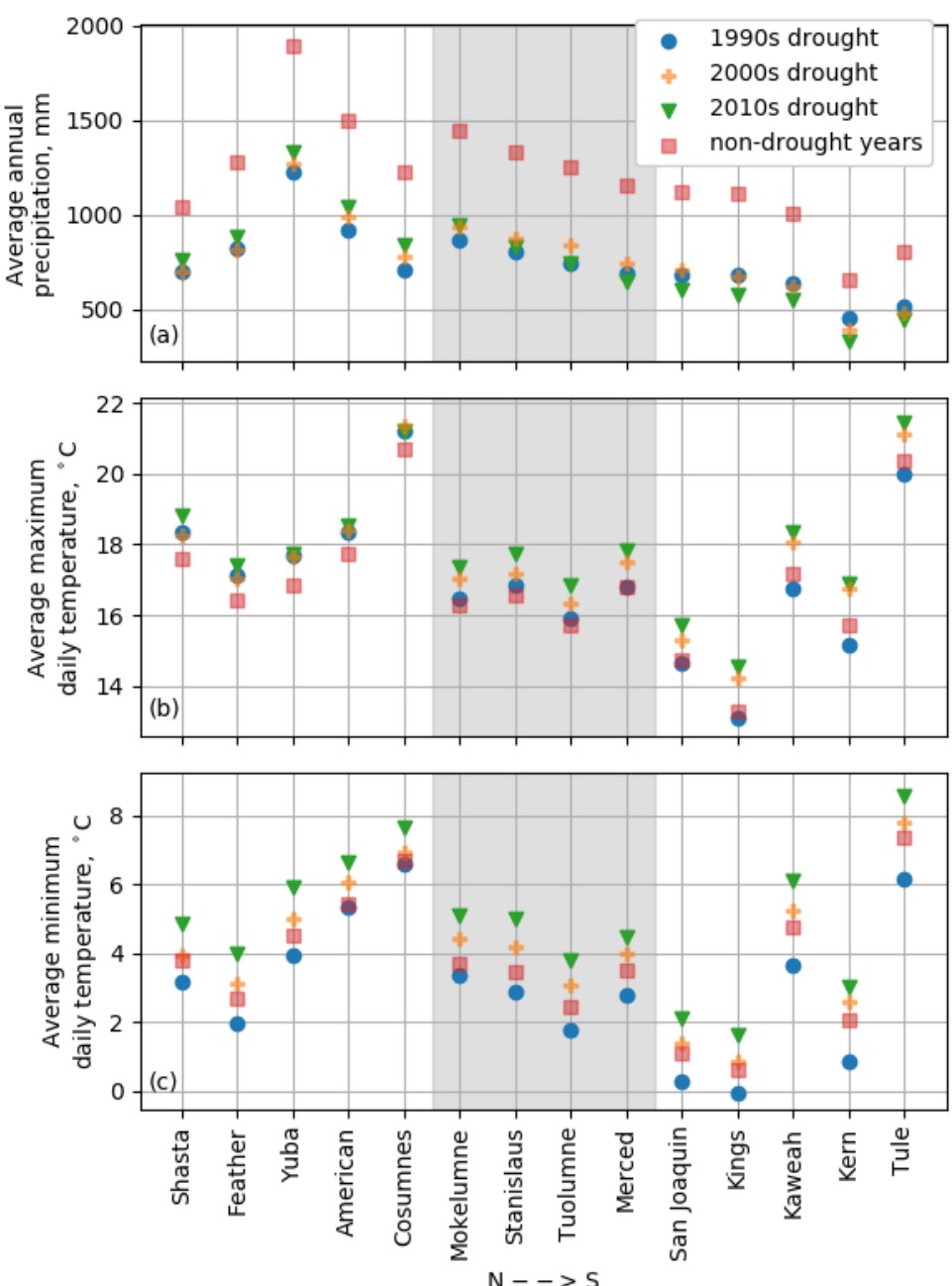

**Figure 3.** Climatic conditions during drought periods. The central Sierra is shaded in gray, with the northern and southern basins to the left and right, respectively. The most recent drought (2010s) was the wettest drought in the northern Sierra, but the driest in the southern Sierra (a). Maximum temperatures only increase in the central and southern Sierra (b), but minimum temperatures increase across the whole study area (c).

northern Sierra, the earlier two droughts were the driest (1990s and 2000s), but but the 2010s drought was driest in the southern Sierra. Thus, droughts in the northern Sierra were progressively wetter with higher minimum temperatures, but droughts in the southern Sierra are progressively drier and hotter (Fig. 3).

## 3.2 Water balance during droughts

After calibration, the *abcd* model showed high performance for simulating runoff in the study basins, with respect to both the Nash-Sutcliffe Efficiency (mean across basins of 0.86, with only the Kern below 0.7) and relative error (mean across basins of 0.16; see Supplement for details). We thus found it suitable for adjusting available water for the annual timestep. Average $\Delta S$ values across all years and basins was -1.2 mm (negative indicating withdrawal), average across years when the subsurface storage was depleted was -29.3 mm, and average across years when it was replenished was 31.6 mm (see Supplement for details). While a handful of years, amounting to 2.7% of all basin years, still lie above the water limit line, the model allowed for stable calibration in all basins of the Fu equation parameter $\omega$ and were thus included in the calibration. Data points falling above the water limit may reflect imperfect calibration in the *abcd* model or be related to uncertainty in the underlying precipitation, ET, and FNF data. These data points may have the effect of increasing the values of $\omega$ in the affected basins, particularly in drought years, which comprise the majority of data points above water limit. However, both drought and non-drought $\omega$ values are on the order of values reported in the literature ($1 \sim 10$; Zhang et al., 2004; Du et al., 2016; Li et al., 2013) for all basins except the Yuba, where extreme energy limitation resulted in very high $\omega$ values (Fig. 4). As the wettest basin in the Sierra Nevada (average annual precipitation of more than 1720 mm), these conditions are consistent with basin climate. However, both $\omega$ values in the Yuba are far outside the normal range, to the point where they are effectively infinity (note that the two lines are indistinguishable in Fig. 4). As a result, we do not consider the direction or magnitude of the shift to carry significance and exclude the basin from further analysis. Since only two $\omega$ values were calculated per basin, it is not possible to directly establish statistical significance of the changes to $\omega$; however, as a baseline, we used a Kolmogorov-Smirnov test to compare changes along each axis (extended aridity index on the x-axis and extended evaporative index on the y-axis) and to $ET/PET$ between drought and non-drought periods. The changes along both Budyko axes between droughts and non-drought periods were significant in all basins to the $\alpha = 0.01$ level ($p < 0.01$), with the exception of change in the evaporative index on the Feather, which was significant to the $\alpha = 0.05$ level ($p = 0.025$). Of the basins that remained in the analysis, the changes in $ET/PET$ were significant to the $p = 0.05$ level in the Merced, San Joaquin, Kaweah, Kern, and Tule and to the $p = 0.01$ level in the rest of the basins (see Supplement for full list of $p$-values).

In general, northern basins saw a shift in favor of runoff (decrease in $\omega$), while the southern basins saw a shift in favor of ET (increase in $\omega$), with the exception of the Cosumnes in the north and the San Joaquin and the Kings in the south (Fig. 4). Note that a shift in favor of ET or runoff does not guarantee that the quantity will increase in absolute terms. Likewise, movement to the right along the same Budyko curve will result in an increase in ET *as a fraction of available water*, but not necessarily in an increase in absolute ET, due to the drop in precipitation during droughts. The Tule and Kern basins in the south see a particularly strong shift in favor of ET (towards a higher $\omega$ value) while the Feather and Mokelumne further north

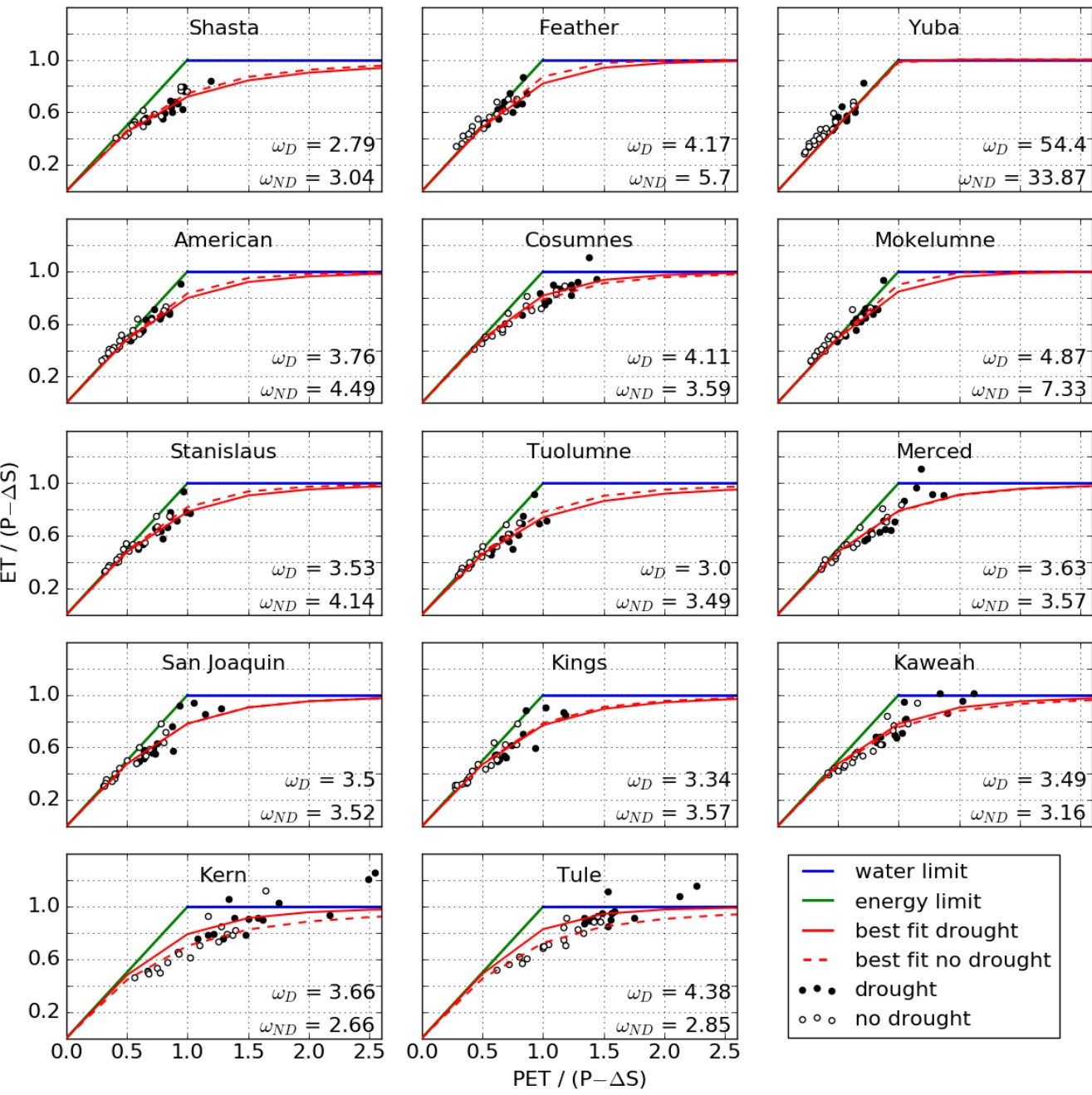

**Figure 4.** Annual (water year) water balances plotted in the extended Budyko framework, with calibrated best-fit lines for drought and non-drought periods. Values of $\omega$ are given for drought ($D$) and non-drought ($ND$) periods.

see the opposite shift (partitioning changes in favor of runoff). In other words, drought may imply increases or decreases in the absolute quantities of ET and runoff.

The absolute changes in ET and runoff due to the partitioning shifts varied both in sign and magnitude, while regime changes were more consistent (Table 1 and Fig. 4). With respect to runoff, the magnitudes of regime-related changes dominate those of partitioning-related changes, with the former at least 10 times higher than the latter in all basins except the Kern and Tule (Table 1 and Fig. 5). This results in an overall drop in runoff across the study area, since runoff regime changes are always negative (Table 1). However, partitioning shifts still account for significant change in the southern Sierra, where regime-related

changes are lower. In the case of ET, changes due to regime shifts still tend to be higher magnitude than partitioning shifts, but not exclusively. As a result, one type of shift can offset the other in basins where they have opposite signs. For example, ET is almost always reduced during droughts from regime shifts alone, but the Feather and Mokelumne would have seen an increase in overall ET if it were not for the curve shift downwards in favor of runoff (regime shift values are positive).

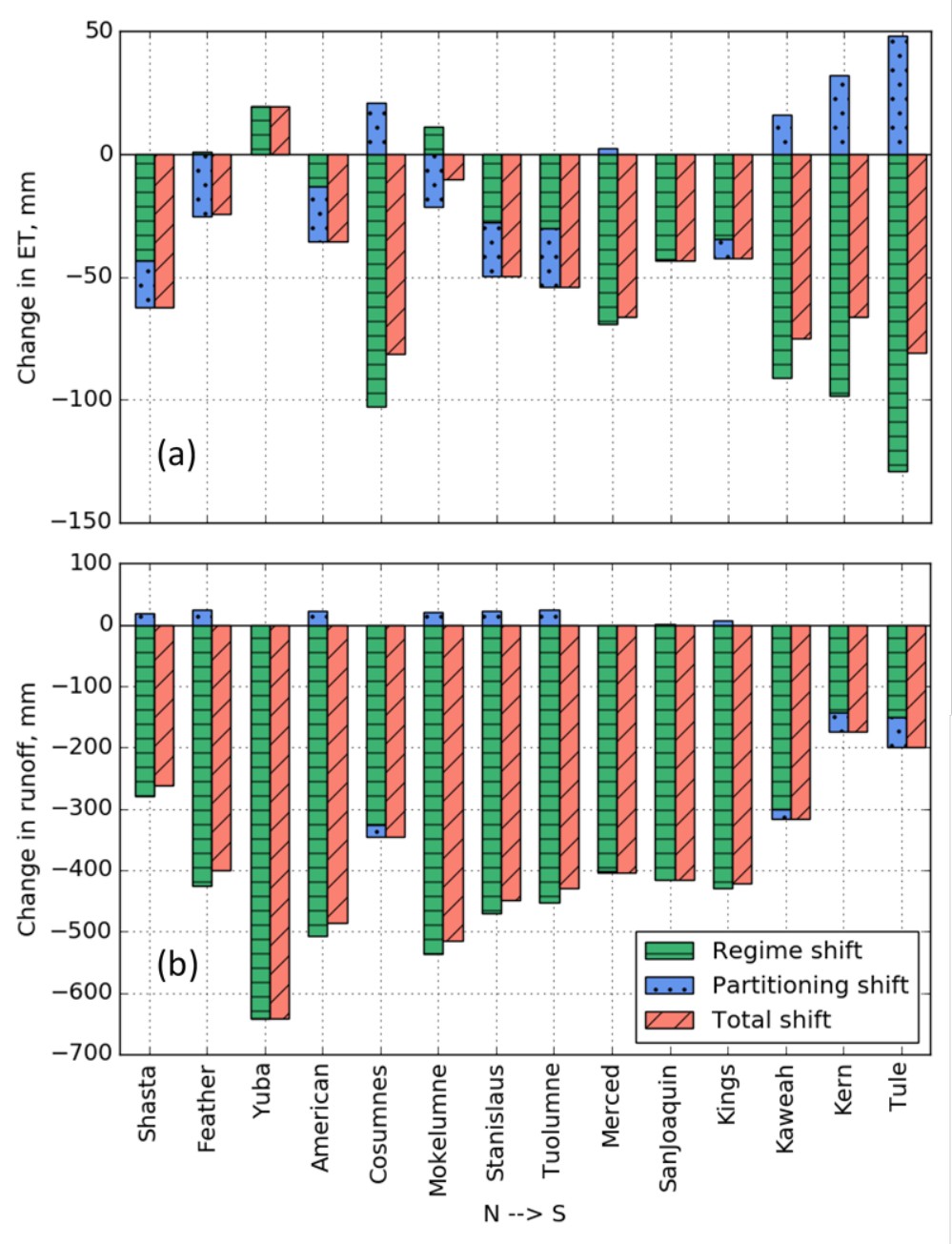

**Figure 5.** Fraction of changes in ET (a) and runoff (b) during droughts that can be attributed to regime vs partitioning shifts.

### 3.3 Drought feedback mechanisms

#### 3.3.1 Catchment aridity

Catchment aridity was higher in basins that saw a shift in favor of ET ($PET/P \geq 0.766$) and vice versa ($PET/P \leq 0.749$). Shift magnitude was highly correlated with average aridity ($r = 0.83$, $p < 0.001$). The threshold dividing the two categories is notable, as $PET/P = 0.76$ has previously been identified as the cutoff between energy-limited water balance regimes and drier regimes, equitant and water-limited (McVicar et al., 2012). Thus, basins where more water than energy is available for evapotranspiration see a shift towards runoff, while those where water and energy availability are more or less equal or where energy is more plentiful see a shift towards ET.

#### 3.3.2 Dry-season baseflow

Baseflow was generally higher in basins that saw a partitioning shift in favor of runoff, with an average baseflow of 14.5 mm in those basins versus 6.9 mm in those that shifted in favor of ET. Most basins with an average baseflow above about 10 mm (a threshold identified manually from the data) shifted in favor of runoff and vice versa (Fig. 6). The only basin that showed a significant departure from other basins displaying similar partitioning behavior was the American, which had relatively low baseflow. Notably, the basins where shifts were the opposite of what would be expected geographically (the Cosumnes shifting towards ET versus the San Joaquin and Kings further south shifting toward runoff) showed the most extreme baseflow values. The Cosumnes had the lowest flows at 2.12 mm and the San Joaquin and Kings had the highest, at 20.9 and 21.9 mm, respectively.

#### 3.3.3 Precipitation phase

Percent of precipitation falling as snow decreased during drought in all basins except the Tule. Northern Sierra Nevada basins saw greater percent decreases than the central and southern Sierra (-2.3%, -1.95%, and -0.52%, respectively), despite the latter having seen greater temperature increases. The northern basins are overall lower elevation, so more area lies in the rain-snow transition where precipitation phase is susceptible to increases in temperature. For the most part, basins with a stronger decrease in percent snow ($> 2.5\%$ decrease, as identified from the data) also saw a decrease in $\omega$ (shift towards runoff; Fig. 6). The Pearson correlation coefficient of $r = 0.62$ ($p < 0.05$) between change in $\omega$ and percent change in snow shows a moderate relationship between the two.

#### 3.3.4 Precipitation excess above 2000 m

Using $P - ET$ above 2000 m as an index for high-elevation runoff and expanding the analysis to the rest of the study site, we find that most basins in the Sierra rely substantially on high-elevation runoff. Nine of the 13 basins analyzed (excluding the Yuba) saw an average high-elevation runoff fraction above 0.33 (those that did not were the Shasta, Feather, American, and Cosumnes). However, overall fraction of runoff from high elevations was not significantly correlated with changes in $\omega$. Instead,

**Table 1.** Change in evapotranspiration and runoff during drought attributable to regime and partitioning shifts

| Basin | Evapotranspiration change, mm | | | Runoff change, mm | | |
|---|---|---|---|---|---|---|
| (N→S) | Total[a] | Regime | Partitioning[b] | Total[a] | Regime | Partitioning[b] |
| Shasta | -58.1 | -47.1 | -11.1 | -287 | -298 | 11.1 |
| Feather | -24.2 | 1.0 | -25.2 | -400 | -425 | 25.2 |
| Yuba[c] | 19.5 | 19.5 | 0.0 | -642 | -642 | 0.0 |
| American | -35.5 | -13.1 | -22.5 | -485 | -507 | 22.5 |
| Cosumnes | -81.4 | -102.6 | 21.2 | -346 | -325 | -21.2 |
| Mokelumne | -10.4 | 11.1 | -21.5 | -515 | -537 | 21.5 |
| Stanislaus | -49.7 | -27.7 | -22.0 | -448 | -470 | 22.0 |
| Tuolumne | -54.3 | -30.0 | -24.2 | -429 | -453 | 24.2 |
| Merced | -66.5 | -69.0 | 2.5 | -405 | -402 | -2.5 |
| San Joaquin | -43.3 | -42.8 | -0.4 | -416 | -416 | 0.4 |
| Kings | -42.3 | -34.8 | -7.5 | -421 | -429 | 7.5 |
| Kaweah | -74.9 | -91.0 | 16.1 | -317 | -301 | -16.1 |
| Kern | -66.2 | -98.4 | 32.2 | -175 | -142 | -32.2 |
| Tule | -80.7 | -129.0 | 48.3 | -199 | -151 | -48.3 |

[a] Totals for each variable are the sum of Regime and Partitioning values.

[b] Partitioning values for evapotranspiration and runoff are the negative of each other.

[c] Since the $\omega$ values in Yuba are both effectively infinite, the partitioning shift has no effect.

we found that changes in high-elevation runoff between drought and non-drought periods was moderately negatively correlated with partitioning shift ($r = -0.55$, $p < 0.05$). In other words, strong decreases in high-elevation runoff during drought were associated with strong shifts in favor of ET and vice versa. Specifically, basins that see a significant decrease ($> 5\%$, again identified from the data) in high-elevation runoff during drought see strong shift towards ET (Tule, Kaweah, Kern). All other basins, including the Cosumnes and Merced, which shifted in favor of ET, saw a positive or small negative percent changes in high-elevation runoff (Fig. 6).

## 4 Discussion

The approach used here has allowed us to distinguish two types of drought-induced shifts, regime and partitioning, based on a new application of existing methods to decompose movement in the Budyko space. To fully explore how this framework can be leveraged to better understand drought implications for the water balance, we present the discussion in three sections. We begin with an explanation of how regime and partitioning shifts primarily relate to climate and basin feedbacks, respectively (Sect. 4.1). Next, we discuss the relative impact of these shifts on absolute values of ET and runoff in the Sierra Nevada during drought (Sect. 4.2). Finally, we offer an interpretation of how partitioning shifts may relate to hydrologic processes by analyzing correlations between shifts and the four basin drought responses enumerated in Sect. 3.3 (Sect. 4.3).

## 4.1 Interpreting regime and partitioning shifts

Due to the nonlinear relationship between the aridity and evaporative indices in the Budyko framework (Fig. 2), both regime
and partitioning shifts result in changes in the precipitation-runoff relationship as observed in other studies (e.g., Avanzi et al.,
2020; Tian et al., 2020; Saft et al., 2016; Petheram et al., 2011). The primary difference, however, is that regime shifts –
movement along the same Budyko curve (Fig. 2b) – are largely reflective of predictable climatic variability during drought,
while partitioning shifts represent a change to a new equilibrium state that cannot be easily forecast a priori. Regime shifts
are almost exclusively controlled by measurable climatic factors through PET (a function of temperature) and precipitation.
Endogenous basin characteristics (i.e., factors influencing available subsurface water storage) are a secondary influence, since
even during drought withdrawals from the subsurface were typically significantly less than precipitation rates (average $\Delta S/P$
for drought years ranged from 0.016 in the Shasta basin to 0.13 in the Tule with an annual maximum of 0.31 in the Tule in 2007;
see Supplement for details). Thus, readily available observations of climate patterns are mostly sufficient to predict regime
shifts and their impact on water resources during drought. Partitioning shifts, on the other hand, are a function of nonlinear and
400 indirect catchment feedbacks to climatic changes during drought. While there is understanding that these mechanisms relate at
least in part to vegetation and subsurface water storage interactions (Avanzi et al., 2020), a relative dearth of data related to both
has so far prevented a full enumeration of these mechanisms and how they interact. This makes the impact of partitioning shifts
on drought water supply largely unpredictable and highlights the need for future research focused on process understanding of
these shifts.

The ability to distinguish these types of shifts while allowing for each to induce nonlinear changes in the water balance is
an advantage of the Budyko framework. Previous studies have used linear models to relate precipitation, Box-Cox transformed
runoff, and a dummy variable to account for drought (Saft et al., 2016; Avanzi et al., 2020). This statistical framing is primarily
concerned with the direct impact of precipitation on runoff. The Budyko framework, however, considers allocation of water
relative to the aridity index, a combination of two major water balance drivers (PET and precipitation), rather than precipitation
alone. Moreover, the Budyko framework governs available water partitioning by physical behavior under limit conditions (when
the aridity index is zero, all water goes to runoff; when the aridity index is one, all water goes to ET). This framework allows
for the possibility that even expected and predictable water balance changes during drought may be nonlinear and that some
shifts observed in other studies may be the result of factors that are not captured in a 2-dimensional precipitation-runoff plane.
This critical difference may explain that though previous studies have observed less runoff than expected without a shift in
relationship (Avanzi et al., 2020; Tian et al., 2020; Saft et al., 2016), most study basins under the Budyko framework show a
shift towards *more* runoff as a fraction of available water than would be expected using non-drought relationships (decrease
in $\omega$; Fig. 4). The direct precipitation-runoff relationship and the Budyko framework are complementary approaches, but the
understanding that water balance shifts during droughts are due to many interacting factors (see Avanzi et al. (2020) and Saft
et al. (2016)) argues for expanding the tools used to analyze this phenomenon. These and new approaches should be the subject
of further study.

## 4.2 Impact of regime and partitioning shifts

Nonlinearities in the relationship between the aridity index, $\omega$, and the evaporative index also mean that regime and partitioning shifts are not equally responsible for changes in ET and runoff during drought (Fig. 5 and Table 1). Regime shifts accounted for at least 75% of runoff reductions across the study area and also dominated changes in absolute ET in most basins. This suggests that most of the observed runoff reduction in a given basin during drought may be predictable if models are expanded from linear precipitation-runoff correlations to include PET. On the other hand, even the relatively small impacts due to partitioning shifts still represent significant volumes of water. For example, partitioning shifts in the Feather River provide 25.2 mm of additional runoff annually during droughts (4.6% of average annual runoff). Over the approximately 9400 km$^2$ basin, this amounts to more than 225 million m$^3$ of water. In the Kern, with an area of approximately 5300 km$^2$, a loss of 32.2 mm yr$^{-1}$ (22% of average annual runoff) due to partitioning shifts translates into nearly 290 million m$^3$.

It is important to note that movement in the Budyko space due to regime shifts do not necessarily indicate whether absolute values of ET and runoff will increase or decrease. Since the extended aridity index ($PET/P - \Delta S$) typically increases during droughts, regime shifts result exclusively in an increase in ET *as a fraction of precipitation*. This results in a decrease to absolute runoff across all basins in the study area, but usually does not translate into an increase in absolute ET (Table 1) due to the available water decreasing significantly during drought. Only in the Feather and Mokelumne basins did ET increase (1 and 11.1 mm respectively), indicating that available water was sufficient to support vegetation. Other than the Yuba, the Feather and Mokelumne basins are the wettest in the Sierra Nevada (average annual precipitation of 1180 and 1290 mm, respectively), while the water availability in the Feather may also be partly supported by the greater groundwater storage in parts of the basin (Avanzi et al., 2020). An increase in ET during droughts has also been observed or predicted in the overall wetter and colder European Alps (Teuling et al., 2013; Mastrotheodoros et al., 2020).

The direction of a partitioning shift, on the other hand, is a direct indicator of the sign of the change in absolute ET or runoff. This is because the partitioning shift relates to change in evaporative index for a given aridity index; in other words, assuming a constant amount of available water. Furthermore, because the derivative of the evaporative index with respect to $\omega$ is nonlinear (see Eq. (3)), the same unit change starting on the higher end of the $\omega$ spectrum will have less impact on the evaporative index than changes on the lower end (Fig. 5). For example, the Feather and Tule see the same magnitude shift in $\omega$ ($|\omega| = 1.52$) but in different directions and starting from different non-drought values ($\omega_{ND} = 5.7$ and 2.85, respectively). In the wetter Feather, the increase in runoff due to partitioning is 25.2 mm, but in the more southern Tule, the decrease in runoff is nearly twice as large at 48.3 mm (Table 1). This is further demonstrated in the Kern and Tule, which had the lowest non-drought $\omega$ values (2.66 and 2.85, respectively) and where runoff was most impacted. This shows that even basins within the same mountain range or region may have high variability in their vulnerability to drought. It further suggests that water agencies that rely on multiple headwater basins (not uncommon in areas like California with highly interconnected water systems), should consider their management strategies on a per-catchment basis.

As with all modeling exercises, some uncertainties exist regarding both regime and partitioning shifts due imperfect underlying data as well as inexact model structure and calibration. Partitioning shifts may be exaggerated by data points that fall

**Table 2.** Summary of basin response mechanisms influencing water partitioning relative to stated threshold

| Basin[a] (N→S) | Average aridity[b] | Average baseflow | Decrease in snow | High-elevation runoff fraction change |
|---|---|---|---|---|
| Threshold: | 0.76 | 10 mm | -2.5% | -0.05 |
| Shasta (Q) | − (Q) | + (Q) | + | + |
| Feather (Q) | − (Q) | + (Q) | − (Q) | + |
| American (Q) | − (Q) | − (ET) | − (Q) | + |
| Cosumnes (ET) | + (ET) | − (ET) | + | + |
| Mokelumne (Q) | − (Q) | + (Q) | − (Q) | + |
| Stanislaus (Q) | − (Q) | + (Q) | − (Q) | + |
| Tuolumne (Q) | − (Q) | + (Q) | + | + |
| Merced (ET) | + (ET) | + (Q) | + | + |
| San Joaquin (Q) | − (Q) | + (Q) | + | + |
| Kings (Q) | − (Q) | + (Q) | + | + |
| Kaweah (ET) | + (ET) | + (Q) | + | − (ET) |
| Kern (ET) | + (ET) | − (ET) | − (Q) | − (ET) |
| Tule (ET) | + (ET) | − (ET) | + | − (ET) |

[a] Effect of shift in $\omega$ is given in parentheses (in favor of Q or ET).

[b] Symbol indicates whether basin characteristic was over (+) or under (−) threshold.

Expected effect, if any, is given in parentheses.

above the water limit line in the Merced, Kaweah, Kern, and Tule basins (see Sect. 3.2 and Fig. 4). Futhermore, as noted in the same section, it was not possible to determine the statistical significance of the difference in the drought and non-drought values of $\omega$, so partitioning shifts in some basins may fall within the range of uncertainty of the data. However, as noted above, even small changes in $\omega$ may have significant implications for water resources, even if a shift is not statistically significant. Regime impacts, meanwhile, may be exaggerated if the Hamon calculations do, in fact, overestimate PET (Sect. 2.2).

## 4.3 Mechanisms of partitioning shifts during drought

In this section, we discuss how the relationships observed between the change in $\omega$ (i.e., movement or observed differences in the Budyko space) and four basin response mechanisms (see Sect. 3.3) may inform our understanding of processes that drive partitioning shifts. These mechanisms are meant to be a non-exhaustive list of plausible, interrelated processes that are related to endogenous basin characteristics dictating the response of the catchment's water balance to drought climate conditions (Troch et al., 2015). It is important to note that this analysis does not seek to esblish a causal relationship between changes in $\omega$ and specific basin characteristics but rather uses correlation and empirical relationships to examine possible controls on the water balance other than the aridity and evaporative indices. Thus, our analysis is intentionally minimal, with a calculation of correlation coefficients as the most rigorous quantification. We believe these initial investigations can provide the basis for more exhaustive future work aimed at defining a mechanistic framework. This approach follows previous literature that has looked at correlations between $\omega$ and vegetation type (Zhang et al., 2001; Ning et al., 2017, 2019, 2020; Roderick and Farquhar, 2011), topographic features like average slope (Yang et al., 2007, 2009; Ning et al., 2019), and soil characteristics

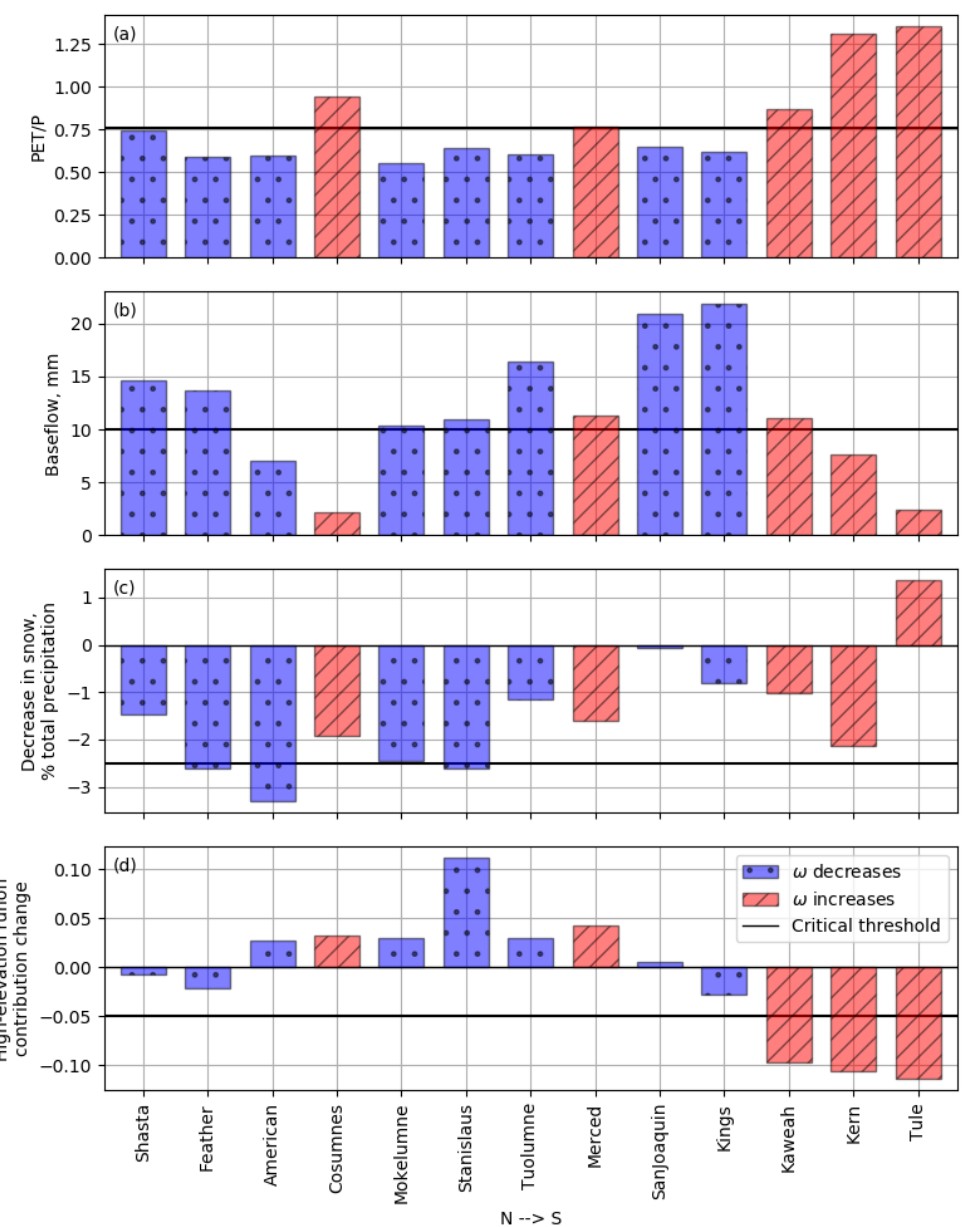

**Figure 6.** Basin responses driving partitioning shifts. Increases in $\omega$ reflect a shift in favor of ET; decreases reflect a shift in favor of runoff.

like infiltration capacity and soil water storage (Ning et al., 2019; Yang et al., 2007). Since the four mechanisms we examine are all fundamentally related to these basin features, our work is broadly consistent with previous literature on the interpretation of this parameter. This discussion includes references to manually identified thresholds in the values used to measure each mechanism, above or below which tendencies in basin behavior can be established (Table 2). These thresholds are not decisive cutoffs (they may be adjusted slightly up or down and still yield the same results) and are meant to add specificity to the discussion of basin behavior rather than to suggest a tipping point in basin response.

Regarding the first metric, aridity, there is a clear pattern in partitioning shifts where the wet catchments (average aridity < 0.76) see a shift in favor of runoff while the arid basins (> 0.76) shift in favor of ET (Fig. 6 and Table 2). This reflects both the greater average aridity of the southern basins as well as the more severe drought conditions (higher temperatures and lower precipitation; Goulden and Bales, 2019). As a lower-elevation basin, the Cosumnes is also more arid and sees a shift toward ET. The high correlation between average PET and shift in $\omega$ suggests that overall climate may predispose basins to a certain drought response through long-term co-evolution of landscapes and climate (Troch et al., 2015). This agrees with previous findings that catchment aridity is a key predictor of shifts in the runoff coefficient (Saft et al., 2016; Tian et al., 2020). Aridity is both a key indicator of catchment climate (Budyko, 1974) as well as being correlated with vegetation and water storage (Saft et al., 2016), both of which also influence the intensity of the feedback cycle between precipitation deficit and vegetation response. Our findings again suggest that dry basins are likely to become drier (i.e., more arid) and that this impact is likely to have a disproportionate impact on runoff compared to wetter basins (Sect. 4.2).

The second metric, amount of dry-season baseflow, provides an estimate of the baseline amount of subsurface storage in a catchment, thus serving as a proxy for a basin's potential for buffering the precipitation deficit with soil storage. Higher baseflows (> 10 mm) were associated with shifts in favor of runoff, reflecting one or more basin mechanisms supporting streamflow during drought (Fig. 6 and Table 2). They may relate to deep groundwater contributions to streams, which are less vulnerable to plant water use, particularly on shorter timeframes, and can thus sustain flows during periods when vegetation is more heavily reliant on near-surface storage. If baseflows are indicative of higher groundwater tables, these soils may become saturated more quickly during a rainfall event, thus leading to saturation-excess runoff (Petheram et al., 2011). The more and higher groundwater tables would make a basin less susceptible to losing this mechanism over large areas during drought (Saft et al., 2016). Finally, areas with higher average baseflow levels are less likely to see storage severely depleted by vegetation over the course of a multi-year drought and are able to continue sustaining streamflow (Rungee et al., 2019). The fact that geographically anomalous basins (Cosumnes, San Joaquin, and Kings) showed the most extreme baseflows suggests that subsurface storage can be a significant factor in basin response, both mitigating and exacerbating drought conditions.

Third, higher temperatures during droughts may induce a shift in precipitation phase from rain to snow, changing the timing of water availability to earlier in the season. Other analyses of the Sierra Nevada water balance during droughts (e.g. Rungee et al., 2019) suggest that snowpack augments plant-accessible subsurface storage by 1) increasing infiltration efficiency, as snowmelt is slow as compared to intense rainfall events, and 2) shortening the length of the dry season by delaying infiltration. As was suggested by Avanzi et al. (2020) and Shao et al. (2012), this implies that shifts from snow to rain may favor runoff rather than ET, at least on the seasonal timescale, since more water is able to runoff or infiltrate to deep groundwater in periods

of low vegetation productivity. Our findings are consistent with this hypothesis: larger changes in percentage of precipitation that fell as snow ($> 2.5\%$) mostly overlap with basins that shift in favor of runoff during droughts (the only exception being the low-elevation Cosumnes basin). Basins where there was little change to snow percentage did not necessarily see a shift in favor of ET, but loss of SWE may be a predictor of greater runoff (Fig. 6 and Table 2).

Finally, the generation of high-elevation runoff, which is more resilient to increases in PET due to overall lower temperatures and sparser vegetation, can help mitigate runoff losses elsewhere in the basin (Goulden and Bales, 2019). Given the orographic effect of the Sierra Nevada, high elevations may also be less susceptible to decreases in precipitation. Our findings on the importance of high-elevation runoff broadly agree with Goulden and Bales (2019), who identified high-elevation runoff as a drought mitigation factor in the Kings River during the 2012–2016 drought. Here, we find that resilient high-elevation runoff is not guaranteed to mitigate drought so much as decreases in high-elevation runoff act to exacerbate drought (again, see (Fig. 6 and Table 2). Both the Merced and Cosumnes basins saw slight increases in high-elevation-runoff fraction during drought, but saw a shift in favor of ET. However, all basins that saw a strong decrease in fractional contribution of high-elevation runoff ($> 0.05$) also saw a shift in favor of ET (Kaweah, Kern, Tule). Thus, high-elevation runoff may not always offset other factors like high aridity and low baseflow, but loss of this important runoff source may shift water allocation towards ET. Alternatively, loss of high-elevation runoff may be correlated with other changes that cause a shift towards ET, such as temperature increases driving increases in ET demand at high elevations or lateral redistribution of precipitation excess from higher elevations to unsaturated soil at lower elevations.

## 5 Conclusions

In this paper, we analyze drought-induced shifts in the water balance of 14 basins in the California Sierra Nevada using the Budyko framework, with the goal of assessing these shifts through an explicitly nonlinear approach. First, we aim to show how a nonlinear framework can identify changes in the precipitation-runoff relationship during droughts. We use a decomposition method to analyze movement in the Budyko space between drought and non-drought periods and identify two distinct types of water balance changes, which we call regime and partitioning shifts. The former is due to changes in the aridity index related to temperature and available water while the latter reflects a change in the Budyko parameter $\omega$.

Second, we aim to distinguish and quantify the impact of these two types of shifts on evapotranspiration (ET) and runoff during drought. We compare the changes in runoff and ET that would be expected from regime shifts alone to the total observed changes. The difference between the two is attributed to partitioning shifts. We find that regime shifts dominate changes in runoff during droughts across basins, but that the total volume of runoff gains or losses due to partitioning shifts are still significant from a water management perspective. Changes in ET are more evenly influenced by both types of shifts. Using this method, the Budyko framework can be leveraged to model more predictable (regime) versus less predictable (partitioning) shifts during droughts while allowing for the possibility that both induce nonlinear changes in the water balance.

Finally, we aim to identify correlations between partitioning shifts – those that cannot be attributed to changes in temperature or water availability – and known basin response mechanisms. A low basin aridity index and high dry-season baseflow as well

as a strong shift from snow to rain and resilient high-elevation runoff during droughts were correlated to greater runoff as a fraction of precipitation during droughts than would be expected without partitioning shifts. These correlations provide evidence that partitioning shifts are related to nonlinear catchment feedback mechanisms between evapotranspiration and storage during droughts and support the use of the Budyko framework as a first-order assessment of drought impacts on water partitioning. Further research is needed to analyze a more comprehensive set of feedback mechanisms and compare the Budyko framework to other nonlinear approaches. By distinguishing between regime and partitioning shifts and quantifying changes in water balance components during droughts, these findings help characterize how water allocations will respond to drought conditions, with implications for natural and human systems in drought-prone regions.

*Data availability.* The data sets generated for this study are available on request to the corresponding author.

*Author contributions.* TM contributed to the study design, data processing, modeling approach, and drafted the manuscript. FA contributed to the modeling approach and manuscript revisions. SG and RB contributed to manuscript revisions. All authors contributed to the article and approved the submitted version.

*Competing interests.* The authors declare that they have no conflict of interest.

*Acknowledgements.* We would like to thank Qin Ma for assistance with the evapotranspiration dataset.

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
