# Peer review of "Table S1: Full-natural flow gauges"

_Hydrology and Earth System Sciences, 2021_

## Author Response (AR1)

Berkeley, California

04 September 2021

Dear Prof. Dr. Harrie-Jan Hendricks Franssen, Editor,

We are pleased to submit revisions for the manuscript hess-2021-55, *Drivers of drought-induced shifts in the water balance through a Budyko approach*, for publication in HESS.

We have extensively revised the manuscript based on comments from the referees and would like to thank you and the referees for the time and consideration in reviewing our manuscript. We welcomed all feedback, and we confirm that all requested changes were feasible.

Please find below our point-by-point replies, including references to changes in the manuscript. Reviewer comments are italicized, and our responses are in plain text. We have also submitted a tracked changes version of the manuscript.

Regards,

Tessa Maurer on behalf of the co-authors

**Reply to Reviewer 1**

*Review on "Drivers of drought-induced shifts in the water balance through a Budyko approach" by Maurer et al. submitted to HESS*

*General comments*

*The authors compiled temperature, potential (PET) and actual (ET) evapotranspiration, precipitation (P) and runoff (Q) data for 14 Californian catchments from a 34-year period with three drought periods to analyze the dependence of ET and Q on P and PET using the Budyko framework. By an innovative approach they quantitatively distinguish drought-induced changes that would be expected within the Budyko framework ("regime changes") from "partitioning" changes that can only be explained by a shift of the curve parameter(s) (in this case, the omega parameter of the Fu equation). They find that regime changes dominate observed changes in ET and Q, while partitioning changes still add non-negligibly to changes especially in some catchments. The topic is relevant to HESS, the methodology sound and original and the results can help understand catchment responses, with the proposed methodology being a potentially useful comparatively simple tool for many other studies in the future. While a number of suggestions for improvement are given below, many of them (hopefully self-explaing which ones) are optional such that from my point of view the manuscript can be accepted after minor revisions. The maybe most relevant single suggestion is avoiding misinterpretations by readers about the degree of novelty of the approach by better acknowledging existing literature on interpreting and decomposing changes in Budyko space (see detailed comments on L84 and L263).*

Public response: Thank you for your comments and, in particular, your point about existing literature. We believe that appropriately situating our work is fundamental to achieving strong science and therefore agree that it should be better laid out for the reader. Please see our responses to your specific comments below.

*Specific and technical comments*

*L42 "wetter, monsoon region in China": something seems to be missing in sentence, check*

Public response: We intended "monsoon" to act as an adjective in this sentence, but understand the confusion. We will change the sentence to "wetter, monsoon-dominated" in the revised manuscript.

Changes to the manuscript: We made this change (line 41)

*L72: be=>been*

Public response: We will incorporate the comments above in the revised manuscript.

Changes to the manuscript: We made this change (line 60)

*L79: A recent study which among others also briefly looks at drought in a Budyko framework: https://doi.org/10.1098/rstb.2019.0524*

Public response: Thank you for bringing this article to our attention. We will incorporate it into our discussion of previous drought assessments using the Budyko framework.

Changes to the manuscript: We included this reference at line 70

*L84: "new framing": This is a bit misleading. Although I'm not aware of your exact methodology (way of decomposing) having been applied to your exact question (distinguishing two directions of drought effect) before, the general idea of using movement along vs. perpendicular to curves in Budyko space to distinguish processes (e.g. climate variability from land-use) is quite widespread, occasionally also including quantification efforts. It would be good to re-check the literature, cite a few examples and adapt the wording. Starting points might be e.g. https://doi.org/10.5194/hess-22-567-2018 (which is already cited but not with reference to the decomposition idea) and doi:10.1029/2011WR011586. It would be good to discuss somewhere how your suggested terms "regime shift" vs. "partitioning shift" relate to already introduced terms in such sources. Both, differences in methodology and scientific reasons, can jusitify your choice of terms (e.g. "climate" vs. "residual" in Jaramillo et al. implies a claim about the causes which it seems you could partly disprove for some catchments); but still it is important for readers that not each paper "reinvents the terminology wheel" without referring to past suggestions.*

Public response: Thank you for bringing this ambiguity to our attention. We did not intend to mislead the reader in suggesting that such an approach is entirely novel, but merely its application to drought conditions. We certainly agree that the papers suggested by the reviewer would provide both more clarity for the reader on the contribution of this paper as well as a sound basis for our approach to droughts. We will clarify the language at both lines 84 and 263, as well as ensure that the abstract, introduction, and discussion are clear on the specific novel contribution of this method to drought contexts.

Changes to the manuscript: We made several changes to clarify this point, including the abstract (lines 14-16); introduction (lines 71-76); methods (lines 251-3); and discussion (lines 378-380).

*L104 PRISM may be a well-known climatology dataset in the US but the description focuses on the interpolation/regression method and does not specify the ultimate source of the original data input to the downscaling / interpolation (e.g. station observations or reanalysis?). Please add a sentence on that so readers all around the world can better judge the potential strengths and weaknesses of the data.*

Public response: Thank you for making this point. Fundamentally, PRISM spatial maps are created based on a regression between digital elevation models (DEM) and a large collection of ground-based precipitation and temperature data, including from the National Weather Service Cooperative Observer Program and Weather Bureau Army Navy stations; U.S. Department of Agriculture National Resource Conservation Service Snow Telemetry (SNOTEL) and snow courses; U.S. Department of Agriculture Forest Service and Bureau of Land Management Remote Automatic Weather Stations; and California Data Exchange Center (CDEC) stations. Depending on the source of the data, different quality control methods were used. Stations are weighted by a variety of factors, including clustering with other stations, distance to pixel, elevation, coastal proximity, and topographic facet. After initial values have been calculated for each pixel, maps are subject to final steps to ensure spatial consistency, such as bound checks on vertical gradients between neighboring cells. We will add a brief description of these details to the revised manuscript.

Changes to the manuscript: Please see the expanded data section (Section 2.2). Lines 103-131 describe the PRISM product in greater detail.

*L105: I guess that inavailability of radiation data was the reason to choose a comparatively crude, less known, semi-empirical PET approach such as Hamon? Here or later e.g. in the discussion, it would be good to comment on the effect it might have had on results.*

Public response: You are correct that the Hamon method was selected since it was usable with the relatively limited spatial data available. We felt that interpolating and distributing very sparse radiation data could lead to even more uncertain estimates, despite the method itself being more sophisticated. We therefore resolved to use a standard approach that has already been applied to environments in the Sierra Nevada (Rungee, Bales, and Goulden 2019). We will include more background on this decision in the revised manuscript.

Changes to the manuscript: We added details on this to the methods (Section 2.2, lines 132-9) and a brief note of implications to the discussion (Section 4.2, line 456)

*L107: Please add one or few sentences on the cornerstones of the ET estimation methodology of Roche et al. 2020. Together with the runoff mentioned in the next sentence, you have everything you need to "close" the water budget (i.e. check for gaps and surpluses in P = ET + Q) and / or quantify the Budyko input parameters P, PET, ET and Q without determining any of them residually, which is good; however, this is only perfectly true if the methodologies to quantify each of them do not implicitly use one or more of the other parameters. As far as I can judge from a quick glance into Roche et al. 2020, this is not a (big) problem here but readers should be put in the position to get a first idea without reading the reference.*

Public response: Thank you for raising this important and valid point. Our data sources are not perfectly independent as the ET regression we used from Roche, et al. uses both NDVI and precipitation. Despite this, we still felt this dataset gave more reasonable ET values for our study area than the calibration based only on NDVI because large portions of the northern Sierra Nevada are significantly wet in winter and including precipitation as a predictor improves ET estimates in such regions (see Roche et al. 2020 for a discussion on this). We have addressed these uncertainties previously (Avanzi et al. 2020), and we found in this study that estimates of the four water balance components tally with expectations and previous work (Avanzi et al. 2020; Rungee, Bales, and Goulden 2019; Roche, Goulden, and Bales 2018; M. L. Goulden et al. 2012; Michael L. Goulden and Bales 2014). This approach is, thus, comparatively established at this stage and provides among the best data-driven estimates available for the region. We will add these details and discussion of the implication to the revised manuscript.

Changes to the manuscript: Again, please see the data section (Section 2.2), specifically, lines 140-159.

*L113: To build further on the comment before, it would be good to report (here, results section or supplement) how large the needed corrections to P were and how much they differed between basins, to give an idea of the overall quality of the dataset - or rather, it's weakest (most assumption-dependent) parameter, which might actually have been ET rather than P.*

Public response: Thank you for making this point. The largest (i.e., highest magnitude) adjustment factor for precipitation was 85.7 mm in Shasta, which also had the largest adjustment as a percentage of long-term average precipitation (7.6%). Wetter basins tended to have higher adjustment factors. (The minimum adjustment factor was in the Stanislaus, at 2.35 mm and 0.3% of average annual precipitation). We will add the full set of adjustment factors, in depth and as a percentage of precipitation, to the supplement.

Changes to the manuscript: We added a summary of these adjustments to the methods (Section 2.2, lines 130-1) and a full table in the Supplement (Table S1). Please note the slight miscalculation of percentage values in the public response was corrected in the revision, but changes did not meaningfully affect the results.

*L119: Mention both PET/P and ET/P consistently as symbols, in words, or both.*

Public response: We will make this revision in the revised manuscript.

Changes to the manuscript: We made this change; see, e.g. line 187.

*L123: (Du et al., 2016) => Du et al. (2016). Same at L126 for Thomas and possibly more places.*

Public response: We will make this correction in the revised manuscript.

Changes to the manuscript: We made these corrections (lines 201 and 212)

*L125-133: Difficult to follow. Consider rewording and/or showing the equation(s), if needed in the supplement.*

Public response: Thank you for bringing this to our attention. We will revise the section to include the equations for the *abcd* model so the description is more concrete.

Changes to the manuscript: We made these changes and additions to the main text (Section 2.3, lines 214-234) and a table of resulting ΔS values to the Supplement (Table S6).

*L152: Remain consistent about writing omega as a symbol or a word.*

Public response: We will make this revision in the revised manuscript.

Changes to the manuscript: We made this change to the manuscript to exclusively use the symbol $\omega$ (see, e.g., line 250.

*L156-163 and Figure 2: The description in the text at the end of section 2.3 and the graphical description in Figure 2 b do not seem to match. I believe the text is "correct" in the sense that the regime shift is consistent with its definition ("what would be expected according to Budyko/Fu") and the partitioning shift is the rest such that both add up to the total observed shift in ET/(P-deltaS). However for the Figure to match this, the vertical blue "partitioning shift" arrow would need to start near the tip (not the foot) of the red arrow / near the centroid of the + symbols, and its tip and the triangles (which are not to refrred to in the text, I think it should be the true observed data of the drought years?) should be further to the (upper) right on the omega=3 line. The difference between these two ways of illustration matters because the distance between the two Fu lines changes with aridity index.*

Public response: Thank you for pointing out the lack of clarity here. In the conceptual figure, we were trying to distinguish the changes due to one type of shift specifically. The triangles therefore represent the hypothetical points that would be observed if only a partitioning shift had occurred. The summation of these changes would be off to the upper right and is what we see in the observed data. We will clarify this point and also add a fourth cluster of points on the conceptual figure to represent the true observed data of the drought years, as the reviewer said.

Changes to the manuscript: We made these edits to Figure 2b and clarified the text in the methods (Section 2.3, lines 254-64)

*Figure 2: Compute more nodes of the Fu equation to make the lines smoother*

Public response: We will make this change in the revised manuscript.

Changes to the manuscript: We made this change to Figure 2.

*L167: with respect \*to\* runoff?*

Public response: We will make this correction in the revised manuscript.

Changes to the manuscript: We made this change (line 267).

*L169: How did "amount of available storage" and the methodology used to estimate it relate to the deltaS values and abcd model used to estimate it earlier?*

Public response: Thank you for asking this; we agree that the phrase "amount of available storage" is vague. In the cited studies, it refers to plant-accessible water storage, in other words, the water that is available to buffer ET against precipitation deficits. The cited papers quantify this value in different ways, but it is conceptually the same as the value we estimate using the *abcd* model. We will change the wording of the phrase to "amount of plant-accessible storage (here, the value estimated as ΔS)" in the revised manuscript.

Changes to the manuscript: We made this change (lines 269-70).

*L175: is estimated \*from\* average...?*

Public response: We will make this correction in the revised manuscript.

Changes to the manuscript: We made this change (line 275).

*L192-193: Unclear: If you refer to changes between droughts (as opposed to between drought and non-drought), then why is there only one difference value per basin given although there were threee drought periods?*

Public response: Thank you for bringing this lack of clarity to our attention. We are reporting average differences for each regions between all drought periods collectively and all non-drought periods collectively. We see how this is confusing when compared with Fig. 3 and will clarify this point in the revised manuscript.

Changes to the manuscript: We clarified this language (line 291).

*L201 / S2: How can a relative error still have units of mm? In case of doubt, specify relative to what / briefly explain the methodology.*

Public response: This was an error on our part; it should be a unitless number, calculated as the summation of error divided by the sum of the observed values. We will correct this in the text and supplement.

Changes to the manuscript: We made this correction in the main text (lines 301-3) and in the Supplement (Table S5). Please also note in both places that a small error in the calculation of the Nash-Sutcliffe Efficiency (not dropping the initial spin-up year) was corrected from the original submission. This slightly improved results.

*L202-203: Were these years excluded from the calibration? Not that I'd like to suggest to do so, it's just that the curent wording almost seems to suggest so.*

Public response: Thank you for asking about this; those years were not excluded from the calibration, so we appreciate knowing about that lack of clarity. We will explicitly state that they were included in the revised manuscript.

Changes to the manuscript: We made this clarification (line 307).

*Figure 5: If regime shifts and partitioning shifty behave strictly additive (without any nonlinear/interactive terms), which it looks like and would be consistent with the methodology description near L163, wouldn't it be more intuitive to use stacked columns? E.g. plotting partitioning shift on top of regime shift - if they have the same sign, the total column length is the total shift, if not the resulting total shift could still be a point marker within the column?*

Public response: This is a good suggestion. Our thinking in plotting them separately was so the reader could quickly distinguish the basins where they shared the same sign from those where they did not, but we believe the reviewer's suggestion may make that even easier. We will test this option during manuscript revisions and incorporate it assuming it meets this need.

Changes to the manuscript: We edited Figure 5 to stack the columns for regime and partitioning shifts.

*Sect. 3.2 in general: While excessive, or rather wrongly interpreted, significance testing is meanwhile disputed (e.g. https://www.nature.com/articles/d41586-019-00857-9), could you think of a simple methodology to roughly transfer what is said from the K-M-tests about changes in the two indices (L211) to the importance comparison between regime and partitioning shifts? While the text qualitytively already tries to convey this message, inspection of the point clouds in Fig. 4 seems to suggest even more that only few catchments (maybe only Kaweah, Kern and Tule) saw a "significant" partitioning shift, while the shifts in all other catchments might be within the range of uncertainty indicated by the scatter of annual data, and thus statistically indistinguishable from "pure regime shifts". Maybe a simple way to try to do this could be to compare the partitioning shift to what would have been significant at p=0.01 or 0.05 in the total ET/PET shifts. A more complex way could be a Monte Carlo type approach where years are randomly removed from the drought / non-drought subset, but maybe this would be overdoing it.*

Public response: Thank you for raising this point; we agree that the implications of *p*-values and statistical significance should be better articulated in the literature. We further agree that we can clarify our use of the K-S tests and their implications. Since only two *omega* values were calculated per basin, it was not possible to directly establish significance with regards to the partitioning shifts. Instead, we used the K-S tests to determine if there was, as a baseline, observable movement along each axis. We found that shifts in all basins and along both axes were significant to the *alpha*=0.01 level with the exception of the ET/P – deltaS shift in the Feather, which was significant to the *alpha*=0.05 level (see Table S4 in the Supplement). We believe that this application of the K-S tests as a simple way to compare the distributions of each value (ET/P – deltaS and PET / P – deltaS) during drought and non-drought periods was appropriate. However, we did not mean to imply that the significance of these shifts along each axis are equivalent to significance in the partitioning shift. We will clarify this point in the revised manuscript. We will also calculate and report the results of a K-S test for the shift in ET/PET values for each basin, again recognizing that this is not equivalent to a significance test for the partitioning shift. We will further re-evaluate the language in Section 3.2 to ensure that we do not misstate the implications of significance to a given *p*-value (as described in the article cited by the reviewer).

Changes to the manuscript: We updated the text to clarify the implications of the *p*-values we present (lines 316-9 and 452-4) and included the results of a K-S test for the shift in ET/PET (lines 321-3 and Supplement Table S7). Please note a calculation error for K-S results in the Shasta basin was corrected from the original manuscript submission. The change did not meaningfully affect the results.

*L246: "Tule. northern..." => "Tule. Northern..."*

Public response: We will make this correction in the revised manuscript.

Changes to the manuscript: We made this correction (line 359).

*L263: See comment on L84.*

Public response: Please see our response to comment on L84.

Changes to the manuscript: See our response to the comment on L84.

*L2278: 10 times less: Is this mentioned anywhere in the results section or at least supplement? Sorry if I overlooked it, but it seems to come a bit out of nowhere here.*

Public response: Thank you for pointing out an oversight on our part in not including the underlying data. This is based on the annual estimates of change in storage from the *abcd* model and the annual precipitation from PRISM. To avoid a large table with thirty years of data for each basin, we will add a table to the supplement with the maximum ratio of subsurface storage change to annual precipitation for each in basin.

Changes to the manuscript: We clarified these values in the text (lines 393-5) and included a new table in the Supplement (Table S2).

*L353: "become drier" - specify in which sense (e.g. less runoff?)*

Public response: We agree this is unclear; in this case, "drier" means more arid (as measured by the aridity index). We will make this change in the revised manuscript.

Changes to the manuscript: We clarified this point (line 484).

*Figure 6 and Table 2: Maybe I overlooked something but other than for the aridity index threshold, which was explained and discussed at L233, the thresholds for the other three parameters are poorly or not connected to the manuscript text (both in terms of explaining how they were determined and of discussing their implications).*

Public response: The thresholds discussed were identified manually from the data. They were meant to serve as estimates and are somewhat subjective. For example, we erred toward selecting round numbers, but in most cases, they can be changed up or down slightly and will give the same results in terms of classifying basin behavior. In the case of the aridity index, the threshold we identified happened to coincide with those identified independently in existing literature. In the revised manuscript, we will adjust in the language in Sections 3.3 and 4.3 to specify that the thresholds are estimated in order to identify *tendencies* in basin behavior, not hard-and-fast cutoffs. In Section 4.3, we will make explicit reference to the threshold values to better connect them to our discussion of basin behavior.

Changes to the manuscript: We made several changes to the results and discussion sections to clarify our use of thresholds and better integrate them into the text. Please see lines 351-2 and 363 in the methods section; lines 471-6, 487-9, 504-7, and 512-16 in the discussion; and edits to the header of Table 2 to better align with the text.

**Reply to Reviewer 2**

*This paper investigates how the partitioning of precipitation into streamflow and evapotranspiration in 14 catchments in California responds differently to drought versus normal meteorological conditions. The main analysis these catchments undergo is based on the Budyko framework; the response of catchments along a calibrated parametric Budyko curve are assumed to be caused by climate, whereas other movements purely in the vertical direction (i.e. E/P or Q/P) that not follow the Budyko curve are assumed to be caused by a change in the hydrological functioning of the catchment. The results suggest that most runoff changes in catchments are caused by "predictable" shifts along the Budyko curve, but substantial effects of regime shift changes are also observed in many of the catchments. These results are further analyzed by examining the correlation between partitioning shifts catchment properties; low aridity index, high baseflow, shift from snow towards rainfall, and the resilience of high-elevation runoff correlate to increased runoff as a fraction of precipitation during droughts.*

*Better understanding the effects of drought on the precipitation partitioning across catchments is a relevant research goal. The use of the Budyko framework to study these changes has been widely applied for other aspects of the hydrological cycle (e.g. human versus climatic effects).*

*While the purpose of the study appears suitable and relevant for HESS, and the methods are largely similar to those widely applied elsewhere in hydrology, I have some questions and comments about the paper that would be good to address*

*The paper distinguishes between "regime" shifts, which result from changes in the aridity index along the same Budyko curve, and "partitioning shifts", which imply a change in the Budyko calibration parameter and thus to the relationships between evaporative demand, precipitation, and ET that govern partitioning of available water. However, what is the physical basis for assuming this? The Budyko framework is developed for characterizing long-term water balances, without any clear theory or evidence that the curve is also appropriate to characterize hydrological change of an individual catchments. I understand that many other papers use a similar approach , and it would be unfair to put the burden of proof on you (and not on the dozens of other previous papers), but I struggle to see how application of the framework like this is justified without any clear theoretical or empirical basis that this is a reasonable assumption.*

Public response: We believe the reviewer makes a valid point that there is no proof of causal connection between changes in the Budyko parameter and basin characteristics, but several studies (e.g., Zhang, Dawes, and Walker 2001; Yang et al. 2007; Jaramillo et al. 2018; Ning et al. 2019) show at least a correlative relationship. Like our approach, these are based on empirical observations, we aim to describe and discuss, although we know that full mechanistic explanations are yet to be formulated. We agree that attempting to establish a quantified, causal relationship between changes to the Budyko parameter (i.e., movement or observed differences in the Budyko space) and specific basin characteristics should be approached with caution. Here, we specifically do not intend our discussion of the mechanisms that possibly influence partitioning shifts to be interpreted as supporting a simple causal relationship, but rather to provide empirical evidence. For this reason, we purposely keep analysis in this section to a minimum, with a calculation of correlation coefficients as the most rigorous quantification. We wish to present a non-exhaustive list of plausible, interrelated causes for unpredictable, nonlinear changes in a basin water balance that are consistent with existing literature on basin drought impacts. We believe these initial investigations can provide the basis for more exhaustive future work aimed at defining a mechanistic framework, but also that it is still valuable to show that droughts may lead to a deviation from a catchment's non-drought Budyko curve.

We also agree that adjustments to the Budyko framework outside the original intended application must be carefully considered. For this reason, we coupled the Budyko framework with the *abcd* model to be able to apply it on an annual basis. While this approach is not perfect (see following response), we believe that errors are within the range inherent in any of the other measured data and spatial grids used to characterize the water balance. In terms of spatial scale, previous work (e.g., Bai et al. 2020; Li et al. 2013) that examines the Budyko framework for smaller catchments does show that other physical processes than water demand and availability are more dominant at these scales than at larger ones. However, we believe this is understanding is consistent with our assessment that other mechanisms and basin characteristics influence movement within the Budyko space.

We will clarify the limitations of the Budyko framework in the revised manuscript, as well as the empirical nature of this study and the opportunity for more theoretical research in the future, with specific mention of the potential for the calibration parameter *omega* to characterize the basin. We will further clarify in Section 4.3 that our discussion of catchment feedback mechanisms is intended to serve as a starting point for further modeling efforts that could establish quantified causal connections using different approaches.

Changes to the manuscript: We made edits to clarify the appropriate applications for the original Budyko framework (lines 191-5); better establish how modifications to the Budyko framework have been handled in the literature (lines 196-204); and clarify how our analysis of catchment feedback mechanisms is consistent with previous work (lines 469-71).  We also clarified that our discussion of mechanisms is not intended to imply causality, but is rather a starting point for more rigorous modeling efforts (lines 460-466).

*Half of the catchments seem to violate the conservation of mass (i.e. ET>P+deltaS) in drought conditions. Does this not suggest that there is something off with the estimates on which all conclusions are based?*

Public response: Thank you for bringing up this point. We do not expect that the *abcd* calibration model used to estimate subsurface storage is a perfect model, and underestimations in the level of subsurface withdrawal in a given year would result in $ET > P + \Delta S$.  However, for the vast majority of years, this method was successful in accounting for subsurface storage and bringing the available water within the theoretical bounds of the Budyko framework. Overall, we believe the errors fall within a similar level of uncertainty to that already extant in the spatial datasets and readings of full natural flow also used in the analysis.

Since the errors may also originate in the underlying spatial datasets, we opted not to estimate withdrawal from the subsurface as the residual of the other components of the water balance for which we have spatial data (ET, P, and Q). This method would prevent any water year from falling above the water limit line, but doing so would assign all uncertainty and error to the $\Delta S$ component and also assume that all water in the system is available for plant use (ignoring, for example, percolation to deep groundwater). This decision was also supported by Reviewer 1 (see comment to Line 107).  In light of these concerns, we felt it would be overly simplistic to use the water balance residual. In addition, since as the reviewer notes, most years that fell above the water limit line were drought years, that it would bias the results to discard those years. We note in lines 202-3 that the years that violate the water limit line amount to less than 3% of all basin-water years.

We will provide more detail about our decision-making process when selecting the *abcd* model for estimating subsurface storage and provide more explanation of the sources and expected ranges of uncertainty. We will also discuss the implications of those years violating conservation of mass for the

results of the analysis, including that the current calibrations of *omega* may lead to slight overestimations of the partitioning shifts in basins where data fall above the water limit line.

Changes to the manuscript: We made edits to the methods section to clarify why we selected the *abcd* model over calculating storage as the residual of the other water balance components (lines 207-211). We also edited the results and discussion sections to discuss the uncertainty implications of having years that fall above the water-limit line (lines 309-10 and 451-2).

**Detailed comments**

*L15: would it be possible to say something more precise than this very generic closing part of the statement?*

Public response: We agree this sentence could be more specific. In the revised manuscript, we will specify that this work has relevance for water resources managers (e.g. dam operators, utility companies, and water agencies) to be better able to 1) forecast changes to runoff during droughts based on available climate data and 2) understand under what circumstances and to what extent their forecasts may be less reliable due to nonlinear basin-climate feedbacks. We will further specify that this work is of particular benefit in arid, drought-prone regions like California.

Changes to the manuscript: We made this change to the manuscript (lines 13-16).

*L26-27: I think this statement needs to be backed up by some references that support this is a widely accepted fact. Personally, I am aware of the possibility this is true, this stating it as an almost universal fact seems like a bit of a stretch (to me).*

Public response: Thank you for pointing out this oversight. Several of the papers currently cited in the manuscript, including Saft et al., 2016; Avanzi et al., 2020; Tian et al., 2020; and Potter, Petheram, and Zhang, 2011 have observed a shift in the precipitation-runoff relationship during drought periods in arid regions, including but not limited to California. We will add these citations to this sentence in the revised manuscript.

Changes to the manuscript: This sentence was modified from the original submission, but please see lines 27-35 and related citations.

*L201: How can the relative error have units mm, and are these relative errors calculated adding up over and underestimations of runoff, causing the overall relative error to be small?*

Public response: Thank you for catching this. The label of millimeters was an error on our part; this should be a unitless number. We will correct this in the main text and supplement in the revised manuscript. Further, we double checked the water balance error values, and they did sum both over- and under-estimations. We have calculated the relative error using the absolute value of the residual. Across the basins, most relative errors in flow were less than 15% with a maximum value of 36%; we recognize that the maximum relative error value has increased at this point, but still believe that the results are functional to our scope given the high Nash-Sutcliffe Efficiency values (see Table S3 in the Supplement) and the fact that we did not use this method for all water balance components, but only as a way to decouple $\Delta S$ values from ET and P, which we believe was important to do (see response to the reviewer's second major question).

Changes to the manuscript: We made this correction in the main text (lines 301-3) and in the Supplement (Table S5). Please also note in both places that a small error in the calculation of the Nash-Sutcliffe Efficiency (not dropping the initial spin-up year) was corrected from the original submission. This slightly improved results.

*L201: is the accurate simulation of runoff an assumed indication that the other fluxes (ET and delta S) also are reliable?*

Public response: Thank you for asking this clarifying question. Yes, the *abcd* model is calibrated to streamflow as suggested in Thomas, 1981 and it is assumed the model then performs well for internal variables. We specifically did not attempt to calibrate the *abcd* model to ET because, as noted in the previous responses, our aim was to estimate $\Delta S$ in a way that was decoupled from the other water balance components in the Budyko framework.

Changes to the manuscript: We made edits to the methods section to clarify the calibration procedure and our reasons for selecting it. Lines 228-31 address this comment specifically, while lines 212-234 include more details clarifying the *abcd* model.

**Reply to Dr. Teuling (Reviewer 3)**

*The manuscript by Maurer and co-workers addresses the issue of changes in water balance partitioning during drought. This is a relevant topic that fits well within the scope of HESS. The authors use a novel combination of methods and data, to arrive at the conclusion that not only the position along the Budyko curve changes during drought, but also the Budyko parameter reflecting the catchment functioning. The manuscript is generally well-written and nicely illustrated. However in contrast to reviewer #1, I unfortunately have some serious concerns about the robustness of the results, and the main motivation for the study, that I think need to be addressed. These are discussed in more detail below.*

*In the Introduction, the authors state that "A particular focus is the change or shift in the precipitation-runoff relationship during droughts, which usually results in less observed runoff per unit of precipitation than would be predicted using non-drought relationships" and that "it is not fully understood which hydrologic mechanisms trigger them". I disagree with this statement, and thereby unfortunately with the main motivation for the study. No hydrologist would claim that runoff response to a unit precipitation input should stay the same across different moisture regimes. In fact, it is well known that the runoff response is a strong and highly nonlinear function of catchment storage on short timescales (i.e. Kirchner, WRR, 2009), which is at least in part related to the nonlinear relation between soil moisture and unsaturated hydraulic conductivity (the main understanding of which dates back almost a 100 years). There is no reason why this would not work similar at longer timescales. The questions here is what we actually don't understand about drought and water balance partitioning, and how the proposed method using a highly conceptual model can provide more insight into this. I believe the authors should do a better job here in formulating a research question that truly reflects, and builds on, the current state of knowledge.*

Public response: We thank the reviewer for bringing our attention to this important point on the framing of our work. We agree that we can better clarify how previous work on drought versus non-drought water balances has been presented. In particular, we agree that we can be more explicit in describing which processes and drivers *are* understood in this context versus those that are not. Specifically, we propose the following changes to the framing:

- (A) While we agree with the reviewer's statement that "*No hydrologist would claim that runoff response to a unit precipitation input should stay the same across different moisture regimes. In fact, it is well known that the runoff response is a strong and highly nonlinear function of catchment storage on short timescales (i.e. Kirchner, WRR, 2009), which is at least in part related to the nonlinear relation between soil moisture and unsaturated hydraulic conductivity (the main understanding of which dates back almost a 100 years)*", we note that some catchments do show a consistent response of runoff to precipitation or, in other words, a linear P-Q relationship with no significant shift during drought (Tian et al. 2020; Coron et al. 2012; Vaze et al. 2010; Saft et al. 2016; Avanzi et al. 2020). Thus, shifts in the water balance during droughts are not unexpected nor entirely inexplicable, but they are also not necessarily expected, either. This leaves open the broad motivating question of this as well as much prior work (e.g., Saft et al. 2016; Potter, Petheram, and Zhang 2011; Tian et al. 2020; Avanzi et al. 2020; Alvarez-Garreton et al. 2021): why do only some basins show a change in hydrologic functioning during droughts and what causes shifts in the places they are observed?
- (B) Furthermore, while the nonlinear relationship between runoff and storage is well-established, other drivers of runoff have also been identified as influencing drought versus non-drought water balances. For example, studies have shown the influence of catchment memory (Avanzi et al. 2020; Alvarez-Garreton et al. 2021), the role of vegetation water use (Saft et al. 2016; Avanzi et al. 2020), changes in precipitation seasonality (Van Dijk et al. 2013) and the influence of catchment topography and elevation (Saft et al. 2016; Tian et al. 2020). Thus, the causes of these

water balance changes cannot be ascribed solely to variability with respect to storage, leaving open the question of what the other drivers and nonlinear relationships influencing this shift are. The locations that these other potential drivers are active and how they interact with each other and with storage is not fully understood. This, together with point (A) above, are the primary motivating questions of our work. In the revised manuscript, we will clarify that some well-understood processes such as storage certainly contribute to a shift in the water balance during droughts, but at the same time the evidence that some basins do not see these shifts leaves an overall incomplete picture of processes and relationships at play.

- (C) From a more technical standpoint, much previous literature (e.g. Saft et al. 2016; Avanzi et al. 2020; Tian et al. 2020; Alvarez-Garreton et al. 2021) has assumed a linear response between P and Q and handled deviations from this response as evidence of "shift". (Again, a linear response between P and Q does exist in some areas, so this is not an unreasonable assumption as a baseline). Thus, it is unclear if the "shift" is simply nonlinearity in the P-Q relationship across a variety of climatic conditions that the linearized approach used by these studies fails to capture, or if it is the signature of some catchment processes being more important during dry periods than during wet periods. Using a Budyko approach provides an opportunity here, in that it accounts for ET and thus allows one to explicitly consider the nonlinearity in the P-Q relationship across a variety of climatic conditions. We found that a certain proportion of what others call "shifts" is in fact explained by mere nonlinearity (what we call a regime shift), while there remains a fraction of the original shift that is less predictable *a priori* (a partitioning shift). That portion of the shift is the signature of other processes being more important during dry periods than during wet periods. Thus, we believe the Budyko framework offers an important insight into characterizing shifts in the water balance during droughts and providing further context for previous literature on the subject. In the revised manuscript, we will explain the current ambiguity in the nature of water balance shifts during drought and explicitly address how the nonlinear nature of the Budyko conceptual model can help clarify it.

- (D) Finally, while there is scientific understanding of nonlinear relationships in the water balance, many operational tools do assume "that runoff response to a unit precipitation input should stay the same across different moisture regimes". For example, seasonal streamflow forecasting by the Department of Water Resources in California relies on a linear relationship between historical snowpack runoff and does not consider soil moisture or drought conditions (Harrison and Bales 2016). Thus, questions like clarifying the full range of processes that contribute to these shifts, in what areas they apply, and how they interact not only solicit more basic science, but also represent an urgent need for formulating more resilient water-resources policy in the current and future climate. Providing different or improved modeling options for these agencies could support more reliable water and economic security. We believe this is also important as a motivating factor for this work, and we will include a brief discussion of this in the revised introduction.

Changes to the manuscript: Please see extensive changes to the introduction, in particular lines 27-35 that clarify that shifts in the precipitation-runoff relationship are sometimes, but not always, observed for a given basin and drought; lines 45-59 that clarify that although nonlinear relationships between water balance components are well-documented, no one relationship is known to consistently be the driver of observed shifts and that, furthermore, it is not clear which of multiple physical mechanisms is at play behind these nonlinear relationships; and lines 45-6 and 65-67 that describe the advantages of revisiting this question using the Budyko framework.

*My second concern deals with the validity of the conclusions. This relates both to the quality of the datasets used and the consistency between them, and to the application of a modified Budyko framework.*

*The results rely heavily on the quality of the data used. Unfortunately, the selection of datasets used by the authors raises a number of questions. Firstly, the precipitation data is rescaled to force long-term average water balance closure (L113-114: "Finally, annual precipitation data were adjusted by the long-term average residual of P − ET − Q so total basin storage over the period of record was zero."). This is a highly unusual procedure, because normally P is the term with the smallest relative error. It is unclear how this procedure was implemented exactly, and how big the corrections were. In addition, it creates an inconsistency with the ET data used, which are calculated based on P which is now inconsistent with the P used in the water balance analysis. The authors should show clearly that this procedure is needed, and that its impact is limited. The rescaling might well mask larger errors in other terms, such as ET. While not much information is provided on ET, it seems to be based on statistical modeling of the relation between observed ET, NDVI, and P. The problem here is that observations of ET over forest ecosystems made by eddy covariance often are inconsistent with runoff observations (the "forest evapotranspiration paradox", see Teuling, Vadose Zone J. 17:170031. doi:10.2136/vzj2017.01.0031. I believe the authors should provide more evidence or arguments on why this ET dataset is useful in this context, and how non-forest and snow areas are dealt with.*

*Perhaps my biggest concern is with the runoff data. Little information is provided on these, so I did some searching on the web myself instead. Based on the following document: https://www.waterboards.ca.gov/waterrights/water_issues/programs/bay_delta/california_waterfix/exhibits/docs/petitioners_exhibit/dwr/part2_rebuttal/dwr_1384.pdf, it seems that the unimpaired flows are subject to numerous calculations and assumptions that even differ between the individual basins. This raises the question if this data should be considered a model product or an observational product. My feeling after reading the document is more the former. This is particularly problematic, because any assumptions in the approach that may impact the runoff values differently in normal and drought years, will directly impact the results. The authors should show that the risk for such bias is small, otherwise their main findings might reflect an assumption made in a modeling chain, rather than an observation that tells us something new about how nature works.*

Public response: We thank the reviewer for raising important points on the quality of the data used in this study. We agree that a more detailed discussion of uncertainty is important and useful for this work. We address the reviewer's comments for the precipitation, evapotranspiration, and runoff data below, and we will include these details in the supplementary information of the revised manuscript.

The PRISM precipitation dataset used in this study is the best-quality gridded data available for the Western U.S., with a monthly mean absolute error of 4.7 to 12.6 mm and a potential annual error of ±98.2 mm (Daly et al. 2008). Also, it is arguably the most used precipitation dataset for mountain hydrology in the Western U.S., and as such it represents a benchmark for hydrologic research in this region (see, e.g., Bolger et al. 2011; Abatzoglou, Redmond, and Edwards 2009; Ackerly et al. 2010; Raleigh and Lundquist 2012; Ishida et al. 2017). At the same time, it is well-established that precipitation uncertainty is high in steep, variable terrain with few ground-based measurements, which includes the montane regions of California. Point measurements that form the basis for interpolated gridded data can underestimate precipitation due to undercatch from wind, wetting loss, evaporation, and trace precipitation (Daqing Yang et al. 1999). The snow-dominated elevations of these regions are subject to further uncertainty in accurately measuring solid precipitation, particularly if precipitation gauges are not heated (Rasmussen et al. 2012). PRISM precipitation in the Sierra Nevada can undermeasure individual storms by up to 50% as compared to snow-water equivalent measurements from snow pillows (Lundquist et al. 2015). Adjusting for errors in gauge measurements on which PRISM is based is common practice (Allerup, Madsen, and Vejen 2000; Bales et al. 2009; Ma et al. 2015; Mernild et al. 2015), and such correction procedures are a necessary choice in other mountain regions as well, regardless of the specific precipitation product used (Avanzi et al. 2021). As a result, the precipitation data likely represent one of the least, not most, certain

components of the water balance for this region, which justifies the adjustment procedure to reduce uncertainty in the dataset.

The adjustment procedure allows for reduction of systemic bias in the precipitation data without assuming that all data uncertainty rests in a specific water-balance component. The procedure was predicated on the assumption that long-term storage in the basin is stable. The procedure was as follows: using the annual, basin-wide values for precipitation, evapotranspiration, and full natural flow, we calculated the residual of P–ET–Q. This value represents the annual change in subsurface and deep groundwater storage in the basin (note that this value is *not* the same as ΔS calculated for this study using the *abcd* method, which represents only plant-accessible subsurface water). Next, we calculated the average of these annual residuals, which represents the adjustment factor. This value was subtracted from the annual precipitation, yielding the precipitation values used in this study. Note that by performing this adjustment, the average of the annual residuals of $P_{adj}$ – ET – Q is zero. As noted in the response to Reviewer #1, the highest adjustment factor was in the Shasta basin and represented 7.6% of the long-term average precipitation, a value that is fully in line with expected accuracy of this dataset and at the same time a minor fraction of both precipitation and all other water-budget terms. We will clarify the description of this procedure in Section 2.2 and, also as noted in the response to Reviewer #1, include the full set of adjustment factors, in depth and as a percentage of precipitation, in the supplement.

The ET data used in this study were from previously published methods (Roche et al. 2020). While our adjustment to precipitation does mean that it relies on different precipitation values, the calibration is distinct from our water balance application. Roche et al. 2020 used the data to perform a spatially distributed calibration rather than calculate the basin-scale water balance. We rely on the authors of the ET dataset to have made the most appropriate decisions for the calibration, and further note that assuming a long-term average storage change of zero for these mountain basins with little exploitation of subsurface water resources is appropriate. Thus, our assessment was that the adjustments of the precipitation data do not make them incompatible with use alongside the gridded ET products, any more than they would if an entirely different gridded precipitation dataset had been used as the index to calibrate the ET products.

The major value of the ET dataset in this context is the opportunity to assess water balance changes without determining any inputs residually, since doing so relegate all uncertainties in the data to a single water balance component (see Reviewer #1's comments on line 107). As with precipitation, uncertainties in the ET dataset are related to both the underlying ground-based data as well as the interpolation method, and uncertainties in the tower-based eddy-covariance data plus satellite data used for scaling are estimated to give a modeling uncertainty of between 10-20% for a given pixel (Roche et al. 2020). Absent a systematic bias in the data, the aggregate basin-scale ET estimate should be lower. This gridded ET product was also based on substantial prior work that provides the theoretical grounding for the regression (see, in particular, Goulden et al. 2012, for an important discussion of why statistical approaches to ET are best suited to the Sierra Nevada region). In addition, while we understand the reviewer's valid concern about the inconsistencies between ET from eddy covariance measurements and runoff, this has been shown to be less of a concern in the Sierra Nevada (see Figure 10 in Roche et al. 2020). Further, the linear P-Q relation provided a good match to P-ET for basins with measured streamflow in the Sierra headwaters (see Figure 1 in the Supplement to this comment). Prior work in regions with similar climates and topography have cited eddy covariance as an accurate method for measuring evapotranspiration (Rana and Katerji 2000; Wilson and Baldocchi 2000; Wang et al. 2015). Variations in land cover and vegetation type are accounted for by use of NDVI in the regression, which has been shown to have a

strong relationship with ET in semi-arid landscapes (Roche et al. 2020; Groeneveld et al. 2007). Since the ET maps were developed on an annual basis and there is no permanent snow cover in these regions, precipitation phase (rain versus snow) was not considered in the regression. We will include these details about the regression methods and associated uncertainty in the revised manuscript.

Finally, we agree that full-natural flows (FNF) are an imperfect substitute for true runoff values. At the same time, we note that estimating runoff using FNF is virtually the only way runoff can be included in hydrologic research in California due to the prevalence of human intervention (e.g. dams and diversions) on the majority of major rivers in the state. While we agree that greater consideration should be given to the implications of substituting FNF for runoff, much research exists that leverages FNF in place of runoff (e.g. Guan et al. 2016; Ejeta 2013; Brown and Bauer 2010; He, Russo, and Anderson 2017; Dettinger and

[Figure]

Figure 2. FNF (Q) versus P-ET for two river basins (Paper in preparation, Roche, Wilson & Bales).

[Figure]

Figure 1. P-Q relations for 4 gauged basins in the upper Yuba River basin. Each data point is one water year. (Paper in preparation, Roche, Wilson & Bales). Note the good agreement in Q and P-ET versus P for the North Yuba at Goodyears Bar, and the near alignment for Oregon Canyon. For the latter, increasing precipitation by 6% aligns the two fits. Making adjustments to Q or ET changes the slope of the lines, and does not result in overlaying the two. For the two Strawberry Valley sites, the difference in slopes reflect the known hydropower diversions; and multiplying Q by 2.0 results in the two fits aligning. Making adjustments to P or ET does not result in overlaying the two.

Cayan 2003; Zeff et al. 2021). Since performing a comprehensive assessment of the implications of this substitution would amount to a full separate research project, it is outside the scope of the present study. However, we agree with the reviewer that a more in-depth discussion of the sources of uncertainty in the FNF calculations and the implications of a FNF-for-runoff substitution is merited for this study, and we will include this in the discussion section of the revised manuscript. For the current study, we simply note that FNF matches P-ET at the basin scale (see Figure 2 in the Supplement to this comment).

Regarding uncertainty, the Department of Water Resources report cited by the reviewer (Huang and Kadir 2016) mentions that most of the uncertainty in FNF values is related to evapotranspiration from overfull

banks and natural wetlands (page 9), which we expect to more heavily impact flows through the Central Valley floor and outflows through the Sacramento-San Joaquin Delta, downstream of outlets of the headwater basins used in this study. The report states in Section 5 (page 79): "Upper rim watersheds, located in the foothill and mountain regions of the Sierra Nevada and California Coast Ranges, are relatively undeveloped. Precipitation-runoff processes are assumed to be assumed unchanged from natural condition for a given climate. Therefore, simulated natural outflows from these watersheds should be similar to estimates of unimpaired flows" (Huang and Kadir 2016). This assumption has been validated in prior studies for certain headwater basins in California comparing FNF to P-ET; see, for example, Figure 5 in Bales et al. 2018 and Figure 10 in Roche et al. 2020.

With respect to the assumptions in the calculations of FNF, the California Department of Water Resources calculates unimpaired runoff starting with measured impaired streamflow or estimated change in reservoir storage. Reservoir evaporation, basin water exports, and irrigation diversions are added, while basin imports and irrigation return flows are subtracted. Differences in the specific adjustments in each basin exist because the type of human intervention, quality measured data on the impact of interventions, and information on historical flow regimes vary across basins (Ejeta et al. 2007).

Changes to the manuscript: Please see major revisions to the data section (Section 2.2). Lines 103-131 discuss the calculation of and uncertainties behind PRISM precipitation maps, including the motivation behind the adjustment of basin-scale precipitation values. Lines 140-159 discuss the calculations of and extensive research base behind the ET maps, and lines 160-179 discuss the computation, validation, and uncertainties of full-natural flow data.

*My second main concern deals with the modified Budyko approach. I share the concern expressed by referee #2 that the modified framework has not been sufficiently tested or proven for the current application. Even in case the framework is valid, there is a fundamental difference between the plots with P, and P−ΔS. In the traditional Budyko framework, the aridity index PET/P reflects an external climate forcing that is decoupled from the catchment itself. Here, the Budyko parameter reflects how the catchment partitions precipitation between ET and Q, at a given atmospheric forcing. In the modified formulation, this interpretation is no longer possible because PET/(P−ΔS) on the x-axis now becomes dependent on catchment properties (through ΔS which is affected by optimization that is different for drought and non-drought years). This means that changes along the Budyko curve can no longer be considered as only induced by climate variation, and changes in the Budyko parameter no longer reflect changes in catchment functioning only. This potentially creates a flaw in the interpretation of the drivers of drought-induced shifts in water balance partitioning. The authors should provide convincing evidence or arguments on why the modified Budyko framework can be interpreted in the same way as the traditional framework. I also suggest to use a symbol for the Budyko parameter that is distinctively different from the "w" used in most papers, stressing the fact that they are not the same parameters.*

Public response: We thank the reviewer for raising this question about the modified Budyko approach. We used a modified approach in acknowledgement that the traditional Budyko framework is not intended for the annual timestep (Budyko 1974), but in practice, the change in storage values are small compared to the precipitation values, so changes along the x-axis will largely reflect climate outputs. Across the basins, average annual ΔS values for a given year ranged between 1.5 and 11.6% of the annual precipitation; we will include these values in the supplemental information of the revised manuscript. The method we use in this paper was developed by Du et al. 2016 and validated in arid, headwater montane regions similar to those examined in this study.

Regarding the reviewer's comment *"ΔS which is affected by optimization that is different for drought and non-drought years,"* we wish to further clarify that ΔS is not optimized differently for drought and non-drought years, since the *abcd* model by which those yearly values were determined was run continuously for all years. Only the relationship between all water balance components (available water, P-ΔS; available energy, PET; and water demand, ET) is optimized differently for different periods. Thus, no procedure should create systemic bias in the water balance values based on the year type (drought vs non-drought). We apologize for the confusion on this point and will revise section 2.3 in the Methods section to ensure the distinction is clear.

Regarding parameter symbols for the modified Budyko framework, we are happy to use a different option in the revised manuscript. However, we note that $\omega$ was also used by Du et al. 2016 when they introduced this modified approach, and we believe it is preferable not to introduce more variation and potentially unnecessary confusion in the already fragmented landscape of scientific literature.

Changes to the manuscript: We made several changes in response to this comment. We edited the discussion to make clear that regime shifts are largely, but not exclusively, driven by climate factors (line 389) and mention values of change in storage relative to precipitation (lines 393-5 and Supplement Table S2).

In the methods section, we clarified that the extended Budyko framework using the *abcd* model has been previously validated for a similar climate to our study area (lines 203-4). We further edited the methods section to clarify the optimization procedure for ΔS (line 229). Finally, we note on lines 238-40 the differences between the $\omega$ parameter in the original and extended Budyko formulations as well as our reasons for remaining consistent with existing literature.

**References**

Abatzoglou, John T., Kelly T. Redmond, and Laura M. Edwards. 2009. "Classification of Regional Climate Variability in the State of California." *Journal of Applied Meteorology and Climatology* 48 (8): 1527–41. https://doi.org/10.1175/2009JAMC2062.1.

Ackerly, D. D., S. R. Loarie, W. K. Cornwell, S. B. Weiss, H. Hamilton, R. Branciforte, and N. J.B. Kraft. 2010. "The Geography of Climate Change: Implications for Conservation Biogeography." *Diversity and Distributions* 16 (3): 476–87. https://doi.org/10.1111/j.1472-4642.2010.00654.x.

Allerup, Peter, Henning Madsen, and Flemming Vejen. 2000. "Correction of Precipitation Based on Off-Site Weather Information." *Atmospheric Research* 53 (4): 231–50. https://doi.org/10.1016/S0169-8095(99)00051-4.

Alvarez-Garreton, Camila, Juan Pablo Boisier, René Garreaud, Jan Seibert, and Marc Vis. 2021. "Progressive Water Deficits during Multiyear Droughts in Basins with Long Hydrological Memory in Chile." *Hydrology and Earth System Sciences* 25 (1): 429–46. https://doi.org/10.5194/hess-25-429-2021.

Avanzi, Francesco, Giulia Ercolani, Simone Gabellani, Edoardo Cremonese, Paolo Pogliotti, Gianluca Filippa, Umberto Morra DI Cella, et al. 2021. "Learning about Precipitation Lapse Rates from Snow Course Data Improves Water Balance Modeling." *Hydrology and Earth System Sciences* 25 (4): 2109–31. https://doi.org/10.5194/hess-25-2109-2021.

Avanzi, Francesco, Joseph Rungee, Tessa Maurer, Roger Bales, Qin Ma, Steven Glaser, and Martha Conklin. 2020. "Climate Elasticity of Evapotranspiration Shifts the Water Balance of Mediterranean Climates during Multi-Year Droughts." *Hydrology and Earth System Sciences Earth Science Systems* 24: 4317–37. https://doi.org/10.5194/hess-24-4317-2020.

Bai, Peng, Xiaomang Liu, Dan Zhang, and Changming Liu. 2020. "Estimation of the Budyko Model Parameter for Small Basins in China." *Hydrological Processes*. https://doi.org/10.1002/hyp.13577.

Bales, Roger C., Michael L. Goulden, Carolyn T. Hunsaker, Martha H. Conklin, Peter C. Hartsough, Anthony T. O'Geen, Jan W. Hopmans, and Mohammad Safeeq. 2018. "Mechanisms Controlling the Impact of Multi-Year Drought on Mountain Hydrology." *Scientific Reports* 8 (1): 1–8. https://doi.org/10.1038/s41598-017-19007-0.

Bales, Roger C., Qinghua Guo, Dayong Shen, Joseph R. McConnell, Guoming Du, John F. Burkhart, Vandy B. Spikes, Edward Hanna, and John Cappelen. 2009. "Annual Accumulation for Greenland Updated Using Ice Core Data Developed during 2000-2006 and Analysis of Daily Coastal Meteorological Data." *Journal of Geophysical Research Atmospheres* 114 (6). https://doi.org/10.1029/2008JD011208.

Bolger, Benjamin L., Young Jin Park, Andre J.A. Unger, and Edward A. Sudicky. 2011. "Simulating the Pre-Development Hydrologic Conditions in the San Joaquin Valley, California." *Journal of Hydrology* 411 (3–4): 322–30. https://doi.org/10.1016/j.jhydrol.2011.10.013.

Brown, Larry R., and Marissa L. Bauer. 2010. "Effects of Hydrologic Infrastructure on Flow Regimes of California's Central Valley Rivers: Implications for Fish Populations." *River Research and Applications* 26: 751–65. https://doi.org/10.1002/rra.1293.

Budyko, M.I. 1974. *Climate and Life*. Academic Press, Inc.

Coron, L., V. Andréassian, C. Perrin, J. Lerat, J. Vaze, M. Bourqui, and F. Hendrickx. 2012. "Crash Testing Hydrological Models in Contrasted Climate Conditions: An Experiment on 216 Australian Catchments." *Water Resources Research* 48 (5): 1–17. https://doi.org/10.1029/2011WR011721.

Daly, Christopher, Michael Halbleib, Joseph I. Smith, Wayne P. Gibson, Matthew K. Doggett, George H.

Taylor, Jan Curtis, and Phillip P. Pasteris. 2008. "Physiographically Sensitive Mapping of Climatological Temperature and Precipitation across the Conterminous United States." *International Journal of Climatology* March. https://doi.org/10.1002/joc.1688.

Dettinger, Michael D., and Daniel R. Cayan. 2003. "Interseasonal Covariability of Sierra Nevada Streamflow and San Francisco Bay Salinity." *Journal of Hydrology* 277 (3–4): 164–81. https://doi.org/10.1016/S0022-1694(03)00078-7.

Dijk, Albert I.J.M. Van, Hylke E. Beck, Russell S. Crosbie, Richard A.M. De Jeu, Yi Y. Liu, Geoff M. Podger, Bertrand Timbal, and Neil R. Viney. 2013. "The Millennium Drought in Southeast Australia (2001-2009): Natural and Human Causes and Implications for Water Resources, Ecosystems, Economy, and Society." *Water Resources Research* 49 (2): 1040–57. https://doi.org/10.1002/wrcr.20123.

Du, C., F. Sun, J. Yu, X. Liu, and Y. Chen. 2016. "New Interpretation of the Role of Water Balance in an Extended Budyko Hypothesis in Arid Regions." *Hydrology and Earth System Sciences* 20 (1): 393–409. https://doi.org/10.5194/hess-20-393-2016.

Ejeta, Messele Z. 2013. "Validation of Predicted Meteorological Drought in California Using Analogous Orbital Geometries." *Hydrological Processes* 28: 3703–13.

Ejeta, Messele Z., Sushil K. Arora, Tariq Kadir, and Hongbing Yin. 2007. "California Central Valley Unimpaired Flow Data." Sacramento, CA. https://www.waterboards.ca.gov/waterrights/water_issues/programs/bay_delta/bay_delta_plan/water_quality_control_planning/docs/sjrf_spprtinfo/dwr_2007a.pdf.

Goulden, M. L., R. G. Anderson, R. C. Bales, A. E. Kelly, M. Meadows, and G. C. Winston. 2012. "Evapotranspiration along an Elevation Gradient in California's Sierra Nevada." *Journal of Geophysical Research: Biogeosciences* 117 (3): 1–13. https://doi.org/10.1029/2012JG002027.

Goulden, Michael L., and Roger C. Bales. 2014. "Mountain Runoff Vulnerability to Increased Evapotranspiration with Vegetation Expansion." *Proceedings of the National Academy of Sciences of the United States of America* 111 (39): 14071–75. https://doi.org/10.1073/pnas.1319316111.

Groeneveld, David P., William M. Baugh, John S. Sanderson, and David J. Cooper. 2007. "Annual Groundwater Evapotranspiration Mapped from Single Satellite Scenes." *Journal of Hydrology* 344 (1–2): 146–56. https://doi.org/10.1016/j.jhydrol.2007.07.002.

Guan, Bin, Duane E Waliser, F Martin Ralph, Eric J Fetzer, and Paul J Neiman. 2016. "Hydrometeorological Characteristics of Rain-on-Snow Events Associated with Atmospheric Rivers." *Geophysical Research Letters* 43: 2964–73. https://doi.org/10.1002/2016GL067978.

Harrison, Brent, and Roger Bales. 2016. "Skill Assessment of Water Supply Forecasts for Western Sierra Nevada Watersheds." *Journal of Hydrologic Engineering* 21 (04016002). https://doi.org/10.1061/(ASCE)HE.1943-%0A5584.0001327.

He, Minxue, Mitchel Russo, and Michael Anderson. 2017. "Hydroclimatic Characteristics of the 2012-2015 California Drought from an Operational Perspective." *Climate* 5 (1): 1987–92. https://doi.org/10.3390/cli5010005.

Huang, Guobiao, and Tariq Kadir. 2016. "Estimates of Natural and Unimpaired Flows for the Central Valley of California: Water Years 1922-2014." Sacramento, CA. https://www.waterboards.ca.gov/waterrights/water_issues/programs/bay_delta/california_waterfix/exhibits/docs/petitioners_exhibit/dwr/part2_rebuttal/dwr_1384.pdf.

Ishida, K., M. Gorguner, A. Ercan, T. Trinh, and M. L. Kavvas. 2017. "Trend Analysis of Watershed-Scale Precipitation over Northern California by Means of Dynamically-Downscaled CMIP5 Future Climate Projections." *Science of the Total Environment* 592: 12–24. https://doi.org/10.1016/j.scitotenv.2017.03.086.

Jaramillo, Fernando, Neil Cory, Berit Arheimer, Hjalmar Laudon, Ype Van Der Velde, Thomas B. Hasper, Claudia Teutschbein, and Johan Uddling. 2018. "Dominant Effect of Increasing Forest Biomass on Evapotranspiration: Interpretations of Movement in Budyko Space." *Hydrology and Earth System Sciences* 22 (1): 567–80. https://doi.org/10.5194/hess-22-567-2018.

Li, Dan, Ming Pan, Zhentao Cong, Lu Zhang, and Eric Wood. 2013. "Vegetation Control on Water and Energy Balance within the Budyko Framework." *Water Resources Research* 49 (2): 969–76. https://doi.org/10.1002/wrcr.20107.

Lundquist, Jessica D., Mimi Hughes, Brian Henn, Ethan D. Gutmann, Ben Livneh, Jeff Dozier, and Paul Neiman. 2015. "High-Elevation Precipitation Patterns: Using Snow Measurements to Assess Daily Gridded Datasets across the Sierra Nevada, California." *Journal of Hydrometeorology* 16 (4): 1773–92. https://doi.org/10.1175/JHM-D-15-0019.1.

Ma, Yingzhao, Yinsheng Zhang, Daqing Yang, and Suhaib Bin Farhan. 2015. "Precipitation Bias Variability versus Various Gauges under Different Climatic Conditions over the Third Pole Environment (TPE) Region." *International Journal of Climatology* 35 (7): 1201–11. https://doi.org/https://doi.org/10.1002/joc.4045.

Mernild, Sebastian H, Edward Hanna, Joseph R McConnell, Michael Sigl, Andrew P Beckerman, Jacob C Yde, John Cappelen, Jeppe K Malmros, and Konrad Steffen. 2015. "Greenland Precipitation Trends in a Long-Term Instrumental Climate Context (1890–2012): Evaluation of Coastal and Ice Core Records." *International Journal of Climatology* 35 (2): 303–20. https://doi.org/https://doi.org/10.1002/joc.3986.

Ning, Tingting, Sha Zhou, Feiyang Chang, Hong Shen, Zhi Li, and Wenzhao Liu. 2019. "Interaction of Vegetation, Climate and Topography on Evapotranspiration Modelling at Different Time Scales within the Budyko Framework." *Agricultural and Forest Meteorology* 275 (January): 59–68. https://doi.org/10.1016/j.agrformet.2019.05.001.

Potter, N. J., C. Petheram, and L. Zhang. 2011. "Sensitivity of Streamflow to Rainfall and Temperature in South-Eastern Australia during the Millennium Drought." In *MODSIM 2011 - 19th International Congress on Modelling and Simulation - Sustaining Our Future: Understanding and Living with Uncertainty*, 3636–42.

Raleigh, Mark S, and Jessica D Lundquist. 2012. "Comparing and Combining SWE Estimates from the SNOW-17 Model Using PRISM and SWE Reconstruction." *Water Resources Research* 48 (1): 1–16. https://doi.org/10.1029/2011WR010542.

Rana, G., and N. Katerji. 2000. "Measurement and Estimation of Actual Evapotranspiration in the Field under Mediterranean Climate: A Review." In *European Journal of Agronomy*, 13:125–53. Elsevier. https://doi.org/10.1016/S1161-0301(00)00070-8.

Rasmussen, Roy, Bruce Baker, John Kochendorfer, Tilden Meyers, Scott Landolt, Alexandre P. Fischer, Jenny Black, et al. 2012. "How Well Are We Measuring Snow: The NOAA/FAA/NCAR Winter Precipitation Test Bed." *Bulletin of the American Meteorological Society* 93 (6): 811–29. https://doi.org/10.1175/BAMS-D-11-00052.1.

Roche, James W., Michael L. Goulden, and Roger C. Bales. 2018. "Estimating Evapotranspiration Change Due to Forest Treatment and Fire at the Basin Scale in the Sierra Nevada, California." *Ecohydrology*, no. March: 1–10. https://doi.org/10.1002/eco.1978.

Roche, James W., Qin Ma, Joseph Rungee, and Roger C. Bales. 2020. "Evapotranspiration Mapping for Forest Management in California's Sierra Nevada." *Frontiers for Global Change*. https://doi.org/10.3389/ffgc.2020.00069.

Rungee, Joseph, Roger Bales, and Michael Goulden. 2019. "Evapotranspiration Response to Multiyear Dry Periods in the Semiarid Western United States." *Hydrological Processes* 33 (2): 182–94.

https://doi.org/10.1002/hyp.13322.

Saft, Margarita, Andrew W. Western, Lu Zhang, Murray C. Peel, and Nick J. Potter. 2016. "The Influence of Multiyear Drought on the Annual Rainfall-Runoff Relationship: An Australian Perspective." *Water Resources Research* 51: 2444–63. https://doi.org/:10.1002/ 2014WR015348.

Thomas, Harold A. 1981. "Improved Methods for National Water Assessment, Water Resources Contract: WR15249270." https://doi.org/10.3133/70046351.

Tian, Wei, Peng Bai, Kaiwen Wang, Kang Liang, and Changming Liu. 2020. "Simulating the Change of Precipitation-Runoff Relationship during Drought Years in the Eastern Monsoon Region of China." *Science of the Total Environment* 723: 138172. https://doi.org/10.1016/j.scitotenv.2020.138172.

Vaze, J, D A Post, F H S Chiew, J.-M. Perraud, N R Viney, and J Teng. 2010. "Climate Non-Stationarity – Validity of Calibrated Rainfall–Runoff Models for Use in Climate Change Studies." *Journal of Hydrology* 394 (3): 447–57. https://doi.org/10.1016/j.jhydrol.2010.09.018.

Wang, Shusen, Ming Pan, Qiaozhen Mu, Xiaoying Shi, Jiafu Mao, Christian Brümmer, Rachhpal S. Jassal, Praveena Krishnan, Junhua Li, and T. Andrew Black. 2015. "Comparing Evapotranspiration from Eddy Covariance Measurements, Water Budgets, Remote Sensing, and Land Surface Models over Canada." *Journal of Hydrometeorology* 16 (4): 1540–60. https://doi.org/10.1175/JHM-D-14-0189.1.

Wilson, Kell B., and Dennis D. Baldocchi. 2000. "Seasonal and Interannual Variability of Energy Fluxes over a Broadleaved Temperate Deciduous Forest in North America." *Agricultural and Forest Meteorology* 100 (1): 1–18. https://doi.org/10.1016/S0168-1923(99)00088-X.

Yang, Daqing, Shig Ishida, Barry E. Goodison, and Thilo Gunther. 1999. "Bias Correction of Daily Precipitation Measurements for Greenland." *Journal of Geophysical Research* 104 (D6): 6171–81. https://doi.org/10.1029/1998JD200110.

Yang, Dawen, Fubao Sun, Zhiyu Liu, Zhentao Cong, Guangheng Ni, and Zhidong Lei. 2007. "Analyzing Spatial and Temporal Variability of Annual Water-Energy Balance in Nonhumid Regions of China Using the Budyko Hypothesis." *Water Resources Research* 43 (4): 1–12. https://doi.org/10.1029/2006WR005224.

Zeff, Harrison B., Andrew L. Hamilton, Keyvan Malek, Jonathan D. Herman, Jonathan S. Cohen, Josue Medellin-Azuara, Patrick M. Reed, and Gregory W. Characklis. 2021. "California's Food-Energy-Water System: An Open Source Simulation Model of Adaptive Surface and Groundwater Management in the Central Valley." *Environmental Modelling and Software* 141 (March): 105052. https://doi.org/10.1016/j.envsoft.2021.105052.

Zhang, L., W. R. Dawes, and G. R. Walker. 2001. "Response of Mean Annual Evapotranspiration to Vegetation Changes at Catchment Scale." *Water Resources Research* 37 (3): 701–8. https://doi.org/10.1029/2000WR900325.

---

## Author Response (AR2)

Berkeley, California

07 December 2021

Dear Prof. Dr. Harrie-Jan Hendricks Franssen, Editor,

We are pleased to submit minor revisions for the manuscript hess-2021-55, *Drivers of drought-induced shifts in the water balance through a Budyko approach*, for publication in HESS.

We have revised the introduction and conclusion of the manuscript based on comments from the referees. We appreciated this new round of feedback and would like to thank you and the referees for the time and thoughtfulness in reviewing our manuscript. We confirm that all requested changes were feasible.

Please find below our point-by-point replies, including references to changes in the manuscript. Reviewer comments are italicized, and our responses are in plain text. We have also submitted a tracked changes version of the manuscript. Please note that no changes were made to the supplement, and, as such, no new version was uploaded.

Regards,

Tessa Maurer on behalf of the co-authors

**Reply to Reviewer 2**

*The authors carefully responded to my reviewer comments (and those of the other reviewers), which improved the manuscript, but the same main concerns raised by myself and reviewer #3 seem to largely persist, and I do not see how they can be fundamentally addressed within a further revision of this paper. While the paper still seems overall a useful contribution, I suggest publishing the paper as is.*

Public response: We thank the reviewer for their feedback and the effort in reviewing our original submission and revised version. No changes were made in response to this comment.

**Reply to Dr. Teuling (Reviewer 3)**

*The author's have generally provided a good response to the issues raised in my previous review, and also to the other reviewers' comments (at least this is my impression after a quick read). The reasoning behind model and data choices are now made more clear, and uncertainties are communicated better. Overall I am statisfied with the new version, except for the fact the manuscript would benefit from a more clear description of the research questions or if possible main hypthesis to be tested (in the final paragraph of the Intro), and a slightly more extensive conclusion section that reflects back on the RQs before answering them. The conclusion section now reads somewhat cryptic, and it is good practice to make this a text that can be read in isolation from the rest of the manuscript. And while the RQs are currently mentioned, the use of similar wording earlier on in the Intro (like "In this paper, we argue...") is confusing and should be avoided. This makes it seem like the goal of the paper is to argue, rather than to provide evidence. This sentence could potentially be moved to the last paragraph of the Intro, after the RQs have been introduced. I hope this helps to further focus the text, and improve the impact of the conclusions.*

Public response: We thank Dr. Teuling for his additional feedback and agree that both the research questions and conclusion could be stronger. We have revised unclear phrases in the introduction (line 45) as well as edited the final paragraph of the introduction to explicitly state our hypothesis (lines 77-8) and clarify our research questions (lines 83-7). We aimed to make the questions clearer and avoid any language that is not already generally established in the literature.

We also significantly revised and expanded the conclusion (lines 525-47) to improve clarity and ensure the conclusion may be read separately from the rest of the paper. This included expanding and clarifying our findings (e.g., lines 526-8; 532-3; and 541-4) and including a brief discussion of future research directions (lines 544-5). We specifically aimed to restate our research aims (lines 526-7; 531-2; and 538-9) as we report on our findings.